# Stimuli-Responsive Boron-Based Materials in Drug Delivery

**DOI:** 10.3390/ijms24032757

**Published:** 2023-02-01

**Authors:** Bhaskar C. Das, Parthiban Chokkalingam, Pavithra Masilamani, Srushti Shukla, Sasmita Das

**Affiliations:** 1Arnold and Marie Schwartz College of Pharmacy and Health Sciences, Long Island University, Brooklyn, NY 11201, USA; 2Department of Medicine, Icahn School of Medicine at Mount Sinai, New York, NY 10029, USA; 3Department of Surgery, Weill Cornell Medical College of Cornell University, New York, NY 10065, USA

**Keywords:** boron, stimuli-responsive, drug delivery, BODIPY, boron nitrite, boronic acid

## Abstract

Drug delivery systems, which use components at the nanoscale level as diagnostic tools or to release therapeutic drugs to particular target areas in a regulated manner, are a fast-evolving field of science. The active pharmaceutical substance can be released via the drug delivery system to produce the desired therapeutic effect. The poor bioavailability and irregular plasma drug levels of conventional drug delivery systems (tablets, capsules, syrups, etc.) prevent them from achieving sustained delivery. The entire therapy process may be ineffective without a reliable delivery system. To achieve optimal safety and effectiveness, the drug must also be administered at a precision-controlled rate and the targeted spot. The issues with traditional drug delivery are overcome by the development of stimuli-responsive controlled drug release. Over the past decades, regulated drug delivery has evolved considerably, progressing from large- and nanoscale to smart-controlled drug delivery for several diseases. The current review provides an updated overview of recent developments in the field of stimuli-responsive boron-based materials in drug delivery for various diseases. Boron-containing compounds such as boron nitride, boronic acid, and boron dipyrromethene have been developed as a moving field of research in drug delivery. Due to their ability to achieve precise control over drug release through the response to particular stimuli (pH, light, glutathione, glucose or temperature), stimuli-responsive nanoscale drug delivery systems are attracting a lot of attention. The potential of developing their capabilities to a wide range of nanoscale systems, such as nanoparticles, nanosheets/nanospheres, nanotubes, nanocarriers, microneedles, nanocapsules, hydrogel, nanoassembly, etc., is also addressed and examined. This review also provides overall design principles to include stimuli-responsive boron nanomaterial-based drug delivery systems, which might inspire new concepts and applications.

## 1. Introduction

Significant progress in drug discovery has been made recently, resulting in novel therapeutic candidates with excellent effectiveness and safety for the treatment of many disorders [1,2,3,4,5]. Moreover, when administered through conventional drug formulations, traditional pharmaceuticals, particularly small molecular drugs, follow a non-specific delivery inside the human body, and it may cause unacceptable toxic effects on normal tissues, leading to serious side effects and limiting the therapeutic efficacy [6,7]. Additionally, several other issues such as hydrophobic nature [8,9], the emergence of multidrug resistance [10,11,12], and poor cell/tissue (membrane) penetration prevent potential therapeutic candidates from being implemented in clinical applications [13,14,15]. The drug delivery system strategy intended to reduce the accessibility of healthy tissue or organisms to the drugs and enhance the pharmacokinetic properties of drugs, including such as solubility and plasma clearance, was supported by the results of issues with conventional pharmaceuticals [16,17,18,19,20,21]. By utilizing the enhanced permeability and retention effect, drugs loaded onto nanostructured materials, such as nanoparticles, nanosheets/nanospheres, nanotubes, nanocarriers, microneedles, nanocapsules, hydrogel, nanoassembly, etc., demonstrated promising accumulation at the target tissue, improving the ability to specifically target the cells while reducing the cytotoxicity to healthy cells [22]. As significant progress has been achieved in the design of DDS, one major problem has emerged: poor controlled release characteristics from DDS. This leads to unwanted early release of the drug and exposure to normal tissues, which reduces the therapeutic efficacy [23]. The approach of controlled drug release was implemented into the DDS to accomplish the appropriate concentration of the drugs to specific areas [24]. Various materials have gained a great deal of interest in the initial efforts focused on investigating materials that release drugs in a regulated manner. It is essential for nanoparticles to have higher drug loading efficiency and site-specific release, while effectively promoting drug release in the affected body locations to increase therapeutic efficiency, and the functionalized materials with these specialized properties have quickly emerged as a viable design toolkit for a variety of various innovative drug delivery systems. The encapsulation can be accomplished by encrypting the stimuli-responsive approach into the system, thus providing on-demand responses to exogenous (light, temperature, magnetic field, ultrasound, etc.) or endogenous factors (pH, enzyme, redox, etc.) [25,26,27,28,29,30]. 

Boron-containing compounds such as boron nitride, boronic acid, BODIPY, etc., have been attracted by the scientific community in recent years, and they have developed as new and immersing research in various applications [31,32,33,34,35,36,37,38]. The biological uses of carbon and boron nitride-based systems especially in drug delivery have increased significantly in the past few years among various nanostructured materials [39]. Carbon compounds with alternating B and N atoms instead of C atoms are structural mimics known as BN. Various nanocarbons have been extensively studied for potentially significant biological uses such as cell targeting, imaging, drug release, sensing, etc.; carbon nanotubes and graphene oxide are two examples of the many nanocarbons being investigated [40,41,42,43,44]. However, there are still many challenges to be solved, particularly those associated with their toxicity. Many in vitro and in vivo experiments have proven that these carbon-based compounds are toxic to human cells and some other living biosystems. In contrast, it has been demonstrated that BN materials have lower cytotoxicity and greater biocompatibility than their carbon counterparts [45], although further research will be required. Designing and developing nanomaterials based on the BN framework should be a viable and promising alternative. Two-dimensional hexagonal boron nitride nanosystems have in recent times attracted considerable attention for a wide range of applications, which include nanomaterials, heat capacity, purification process technology, electronic equipment, and catalyst owing to the high surface area, considerable wide band gap, and distinctive optoelectronic properties [46,47,48,49]. Currently, the potential uses of BN nanomaterials have extended to drug delivery [50,51,52]. Owing to the existence of numerous small-scale surface connections, BN nanomaterials have an extremely high surface-to-volume ratio that provides a high drug loading efficiency and may promote effective drug delivery to specific sites. In addition, tunable BN nanomaterials design opens up many options for biological and pharmaceutical applications due to their tunable shape, depth, size, and surface characteristics.

Organoboron molecules (phenyl boronic acid) exhibit tremendous potential in biological applications [53,54,55]. PBA-based materials are growingly being used in biomedical fields, such as stimuli-responsive nanomaterial-targeted delivery of protein, gene, and chemical drugs, and bio-imaging in vitro and in vivo for the recognition of different bioactive molecules such as sialic acid, carbohydrates, adenosine triphosphate (ATP), and dopamine. PBA has been employed as a target component for tumor-targeting drug delivery due to the unique interaction between PBA and sialic acid, which will be abundantly expressed in many types of tumors [56,57]. In addition, PBA’s ability to accept electrons allows for the nucleophilic combination of nanomaterials with electron-donating ions such as oxygen and nitrogen, which can be used to construct controlled drug delivery with high drug loading [58]. Additionally, materials containing PBA exhibit a nucleus-targeting capability, interact with electron-donating units such as amines, generate new boronate ester bonds with diol compounds including sialic acid, carbohydrates, ATP, and dopamine, and can stoichiometrically utilize reactive oxygen species (ROS). Recently, responsive boron nanomaterials have been developed for biomedical applications, such as (i) nanomaterials with PBA that target sialic acid for the delivery and monitoring of drugs that target tumor cells; (ii) saccharide-binding PBA-based nanomaterials used for insulin release in response to glucose, different cell capture and release, and self-healing components; (iii) PBA-based drug carriers with high drug loading efficiency; (iv) ROS-responsive PBA-based nanocarriers for delivery of drugs and ROS sensors; (v) PBA-based drug carriers with ATP sensitivity and ATP sensors; (vi) PBA-based monitoring systems with dopamine responsiveness; (vii) PBA-based nanomaterials for nuclear-targeted delivery of the drug.

Due to their diverse area of applications, 4,4-Difluoro-4-bora-3a,4a-diaza-*s*-indacene, also known as boron-dipyrromethenes, are receiving a lot of attention [59]. They have gained considerable research in sensors, bioimaging, catalysts, laser dyes, solar cell components, photoremovable protecting groups, photodynamic therapy, and fluorescent indicators [60,61,62,63,64,65]. This is brought about by their exceptional qualities, which include less toxicity, comparatively extended excited-state lifetimes, excellent chemical and photochemical durability, photo-stability, large molar extinction coefficients, high fluorescence quantum yields, and artificial photosynthetic systems [59,60,66]. Additionally, the BODIPY-moiety can be simply modified to alter its optical and electronic properties. For instance, as shown in Figure 1, the small structural modification of the BODIPY-core to the azaBODIPY core, namely the substitution of a nitrogen atom for one CH_2_ group, causes a dramatic bathochromic shift of approximately 100 nm. At least one substituent is added to the pyrrole group due to the instability of those exhibited basic framework structures [67]. In the green area (490–570 nm), this chromophore mostly displays strong absorption and emission spectra with minor stokes shifts. Many chemists can fine-tune the characteristics of BODIPY’s absorption and emission properties because different substituents can be functionalized at the 8-positions in the BODIPY core by nucleophilic or electrophilic substitution, C-H activations, and cross-coupling reactions. These have generated numerous original concepts for using BODIPY derivatives to identify biomolecules and provide photodynamic therapies. 

This review aims to provide useful information to the scientific community to design/develop stimuli-responsive boron-based drug delivery systems. We outlined the different approaches systematically according to the various substitution/caging of the boron-based systems. We will be discussing the recent developments and advances in the synthesis of boron-based drug delivery and their current uses in various diseases (Table 1). 

## 2. Types of Drug Delivery Methods

There are numerous ways that drugs can enter the human body (Figure 2). These pathways are typically categorized according to their “initial point” or the place where the drug is delivered. Every approach offers benefits and drawbacks of its own.

### 2.1. Buccal Drug Delivery

The technique of delivering a drug through the buccal mucosa is known as the buccal delivery of drugs (lining of the cheek). Delivery is mainly only possible with chemically synthesized drugs with lipophilic qualities because they can easily penetrate the membrane, even though this delivery technique minimizes first-pass impacts (efficient drug intake and conversion into inert chemicals by the liver). Substances that can adhere to the mucosa are often favored since the buccal approach is frequently utilized for the prolonged delivery of drugs (where the drug is given in a controlled manner over a long period. For buccal delivery, a number of compositions have been created, including pills, creams, gum, and patches [114,115].

### 2.2. Nasal Drug Delivery

The nasal delivery system refers to the administration of drugs through the nasal passages. Typically, drugs delivered by nasal spray are utilized to treat upper respiratory tract localized illnesses. Moreover, this delivery technique can be utilized for systemic distribution of smaller molecular drugs, such as the headache treatment almotriptan, in specific situations (when, for example, quick initiation is necessary). The thin nasal mucosa is responsible for regulating blood, allowing for quick transport to the central blood flow and, like buccal delivery, the avoidance of first-pass metabolism. For nasal delivery of drugs, fluid and (lower usually) powdered forms can be utilized [116,117].

### 2.3. Ocular Drug Delivery

Owing to the eye’s unique architecture and metabolism, drug delivery researchers have discovered it challenging to distribute drugs through the eye. Stable, dynamic, and enzymatic retinal barriers all prevent absorption of the drug through the eye. Drugs can be administered via a variety of different methods to target areas of the eye. By discovering particular efflux and inflow carriers in the eye and altering drugs to target particular carriers, scientists have managed to substantially overcome the difficulties of distributing drugs to ocular regions [118].

### 2.4. Oral Drug Delivery

Due to its non-invasiveness, simplicity, cost-effectiveness, and extremely absorbent qualities of the gastro (GI) tract, oral drug delivery is without a doubt the most well enough and frequently preferred method of drug delivery. In order for oral administration to always be effective, the drug compound’s GI state’s solubility must be evaluated to see if adjustments are needed to increase biocompatibility. Furthermore, oral therapy does not always work well for some types of patients, such as children, the elderly, and people who have memory loss [119].

### 2.5. Pulmonary Drug Delivery

The process of administering a drug by inhaling it via the nose and entering the lungs is known as pulmonary drug delivery. For the treatment of localized lung illness, inhaled drugs are beneficial. Because of the respiratory region’s large adsorbing surface area and highly porous outer layer, pulmonary drug delivery has also lately been investigated as a potential treatment option for systemic disorders. Dietary issues and interpatient metabolic diversity have little impact on pulmonary delivery, which is another benefit [120].

### 2.6. Sublingual Drug Delivery

The process of administering a medication under the mouth so that it can be taken into the bloodstream through the anterior surface of the mouth and the tongue’s floor is known as sublingual drug delivery. Since sublingual uptake is speedy, an immediate commencement of effect is possible. Additionally, this administration method prevents systemic first-pass metabolism. Moreover, it causes disruptions in speaking, eating, and having a drink, which is bad. Additionally, smoking reduces the absorption of the drug and, as a result, effectiveness owing to vascular constriction of the arteries; hence, use in smoking is not really advised [121].

### 2.7. Transdermal Drug Delivery

A drug can be administered systemically using the transdermal drug delivery technique, which involves putting a formulation on healthy skin. The drug first enters the cell membranes, then moves into the inner epidermis and dermis, and then eventually enters the bloodstream through the dermal vascular system. Two major benefits include being non-invasive and appropriate for individuals who are vomiting or asleep [122].

### 2.8. Vaginal/Anal Drug Delivery

When compared to oral ingestion, vaginal/anal drug delivery methods have a quicker beginning of action as well as greater bioavailability. Rectal drugs may have local (such as laxative impacts) or central nervous system effects. Drug delivery through the vagina bypasses first-pass metabolism and seems to be undisturbed by digestive issues. When addressing female health issues and administering hormones, the vaginal approach is considered generally. There are many alternatives for vaginal formulations, including biopolymers, pills, pessaries, and suppositories [123,124].

In this section, we have discussed generally available drug delivery methods. Most of the drug delivery systems are oral-based delivery and compared to oral other types of drug delivery methods have some advantages. The advantages of each method are explained in this section. Our aim is that researchers should focus more on other types of drug delivery systems in the near future that will be a good fit for real-time applications. 

## 3. Types of Materials Used in Drug Delivery

In recent years, different types of materials have been used in drug delivery. The main objectives of drug delivery are: (i) site-specific targeted drug delivery, (ii) high drug loading efficiency, (iii) allowing for precise spatial and temporal control over the release, and (iv) increasing therapeutic activity while lowering harmful side effects. This section examines various types of materials designed/developed for drug delivery [125,126,127,128,129]. 

### 3.1. Polymers

Polymers are a type of material that is especially well suited for use in nano-size drug delivery systems because they provide practically limitless diversity in chemistry, dimensions, and topology [130]. Understanding the framework interactions of polymers is advancing their usefulness. Linear, branching, cross-linked, block, graft, multivalent, dendronized, and star-shaped polymers are only a few of the many different types of polymer topologies that exist [130,131]. Notably, chemical nature (polyester, polyanhydride, polyamide), cargo durability (biodegradable, non-biodegradable), and solubility in water (hydrophilic, hydrophobic) can all have a substantial impact on how well drug delivery vehicles work [130,131]. In other words, the physicochemical properties of the material are determined by the copolymer morphology for the same monomer percentages. The physicochemical characteristics of the vehicles are influenced by polymer design, which also has an impact on drug loading effectiveness, the release of a drug, and bioavailability [132]. Drugs may be covalently bonded to the polymer network or physically confined inside polymeric capsules and frameworks. By enhancing the drug’s active binding capacity, polymer–drug conjugates, which feature a physiologically stable, bio-sensitive polymer–drug linkage, change the pharmacokinetics of the drug. The linker makes sure that the pro-drug is inert in circulation until the right enzyme or pH specifically releases it at the target site. It will be possible to develop new and enhanced polymeric nanoparticle drug delivery systems by combining biological rationale with reducing synthetic chemistries [133]. These chemistries have to be adaptable to industrial-scale development to be available for therapeutic uses. Researchers are still looking into novel biodegradable polymers with more complex three-dimensional frameworks that are more appropriate for regular parenteral delivery [131].

### 3.2. Nanoparticles

Drug delivery systems can generally be applied locally or systemically and have the potential to attach targeted molecules. Both inside and outside of the target cells might be exposed to the drug payload. Shorter drug delivery systems can directly endocytose cells as opposed to bigger drug delivery systems, which can produce high localized drug levels. Nanoparticles are transported from early endosomes to selective endosomes after being taken in by the cell. Although some of the nanoparticles are carried to secondary endosomes or lysosomes, where they might release and act as intracellular drug repositories, other nanoparticles are ejected from the cell [129]. Lower particle sizes allow for greater capillary penetration, easier passage through fenestrations, and subsequently greater cellular uptake. In addition, in situ uptake capacities are 100 nm, and particle size is 15–250 times higher than those of smaller microparticles [134]. Even the blood–brain membrane can be crossed by nanoparticles [135]. Different formulation techniques have been investigated to manage drug delivery system sizes as well as to enhance encapsulated drug and release patterns. Similar to this, adding targeting ligands will raise the drug’s effective concentration at the targeted site [136].

### 3.3. Nanocapsules

Nanocarrier techniques, such as lipids and polymeric nanocapsules, can offer controlled drug release and effective targeting [137,138,139]. Its distribution durability and the important physiological action, specifically, are determined by the composition of the outer layer. Layer-by-layer deposition, surface polymerization, adhesive precipitation, interfacial deposition, and self-assembly processes can also be used to generate nanocapsules [137]. The diameter variation, shell thickness, membrane decomposition, and detergent type are all significant factors. Lipid-based nanocapsules could be altered to target particular tissues or cells through antibody binding and to change capillary integrity by channel implantation. It has been demonstrated that lipid-based nanocapsules retain their payload [140]. However, the usage of lipids may be constrained due to their instability in physiological environments and sensitivities to a wide range of environmental factors, such as temperature and osmotic pressure [141]. Lipid–polymer-associated nanocapsules could be developed to increase the durability of lipid-based nanocapsules. Approaches include encapsulating the liposome with a polyelectrolyte shell, incorporating surface-active polymers to form hybrid extracellular structures, and polymerizing a two-dimensional system in the hydrophobic groups of the membranes [141]. The construction of capsules composed of disulfide cross-linked polymers was another exciting accomplishment [142]. At optimum pH, hydrogen-bonded layered nanomaterials are more stable because of the disulfide bonds. They also make the system liable to disintegration when thiol-disulfide exchange agents are present. Because intracellular proteins such as glutathione will facilitate in vivo capsule deconstruction, such nanocapsules always had the potential to be developed as biodegradable nano-scale drug delivery platforms [143,144].

### 3.4. Nanotubes

For some purposes, nanotubes, which resemble small reusable straws, have benefits over spherical nanoparticles [145]. Various substances and compounds, varying in size from tiny molecules to proteins, can be stuffed into their spacious interior chambers [146,147]. Certain types of nanotubes can be selectively manipulated to encapsulate particular drugs internally and to avoid an immune response outwardly because the inner and exterior surfaces of these nanotubes are dissimilar [147]. Lastly, loading is made exceptionally easy by the accessible design of nanotubes. Nanotubes can be produced using a variety of materials and methods, such as controlled coating and self-assembly. Instances contain pattern nanotubes, cyclic peptide nanotubes, and heterocyclic nanotubes [145]. Metallic nanotubes can be made using electrodeposition, polymeric nanotubes can be made through in-pore polymerization, while inorganic nanotubes can be made through sol–gel technology. The most flexible method for creating nanotubes is the template method. The component (polymer, silica, metals, or carbon) is deposited into the spherical pores of a solid surface to form the nanotube [148]. The template’s size determines the outside diameter, whereas the deposition time determines the inner diameter. This technique can also be used to create composite nanostructures such as divided nanowires and cylindrical tubular structures [149,150]. The control of the caps to regulate drug delivery will now be the main subject of interest in nanotube development. By adjusting the magnetic properties, nanotube rate, and stabbing period, penetration effectiveness can be modified [151,152]. The lack of cytotoxicity that carbon nanotubes exhibit at lower nanotube levels (10 µM) may be related to the nanoscale penetration of these materials, which causes less disruption than other, better mechanical delivery techniques [153]. Nanotube spearing is a significantly more effective propagation technique than conventional magnetoreception or endocytosis because the DNA cargo does not depend on lysosomal release and can also be delivered straight into the nucleus [154]. In addition, compared to other membrane-penetrating methods, nanotube diving is significantly better suited to high-throughput biochemical investigations, and it operates at its best at only 100 fm nanotubes [153].

### 3.5. Nanogels

Hydrogel frameworks are suitable for use in drug release because they have good bioavailability and adjustable characteristics as well as help to avoid payload agglomeration [155]. The ability to be synthesized without the use of drugs, minimizing the risk of drug deactivation, perhaps is hydrogels’ greatest benefit as drug transporters [156]. Usually, the drug is eventually loaded using non-covalent self-assembly techniques. Incorporating charged and hydrophobic macromolecules is possible in hydrogel frameworks, which are ultimately defined by the physical characteristics of the polymers that make them up [127]. Nanoscale hydrogels, also known as nanogels, are simple to make and have a good drug-loading capability. In the presence of water, hydrogels, which are three-dimensional cross-linked polymer networks, swell [157]. They can be made to react to a range of physiological triggers, such as temperature, pH, and ionic strength. Incorporating both physical and chemical cross-linking components, hybrid polymerizable nanogels have been developed [157]. Nanogel particles have a high surface-to-volume ratio, a microheterogeneous shape, and a compact size, which combine the characteristics of gels and colloids. Polymeric nanogel carriers exhibit prolonged durability, controlled release, minimal side effects, and resistance to enzymatic degradation [158,159]. It has recently been demonstrated that inverted emulsion photo-polymerization can be used to generate well-defined, durable, cross-linked, amphiphilic hydrogel nanoparticles [160]. By altering the polymerization parameters, the colloidal substance’s size, which can include hydrophobic drugs, can be tuned. The hydrophilic corona reduces agglomeration, protein adsorption, and immunogenic response, whereas the hydrophobic core dissolves lipophilic substances [160]. This technology may lead to the development of a novel type of colloidal nanoscale drug release platform because it is responsive to numerous amphiphilic precursors.

### 3.6. Dendrimers

Dendrimer architecture is highly controllable, resulting in well-defined form, size, branching length, thickness, and surface functionality [125,161]. Thus, dendrimers are desirable candidates for nano-scale drug-delivery platforms. The drug payload may be chemically or physically bound to the surface of the dendrimer. The increased density of exo-presented surface functional groups on dendrimers makes them more targetable and biocompatible [125]. The continuous click strategy or the convergent “Lego” strategy can be used to synthesize dendrimers, which has recently been made easier [162,163]. Both methods result in byproducts that are harmless to the environment and allow for simple purification. Chemotherapeutic drugs such as cisplatin [164], methotrexate [165], and 5-fluorouracil [166] have been the main focus of dendrimer delivery drugs because they offer slower release, better accumulation in tumor cells, and reduced toxic effect compared to free drugs, especially whenever the dendrimer is PEGylated. Dendrimer permeability rises with dendrimer size, and ester-terminated dendrimers are much more accessible than their amino-terminated equivalents for a given surface-area-to-volume ratio [167]. “Bow-tie” designs that covalently bind a drug-loaded dendron to a PEGylated, dissolving dendron and mixed dendrimer-based microcapsules that offer a dual releasing strategy are examples of innovative dendrimer platforms for uses as nano-scale drug delivery systems. Dendrimers can serve as naturally active anticancer, antiviral, and antimicrobial agents as well as penetration enhancers that can enhance oral and epidermal drug delivery, and they can act as drug delivery carriers [168,169,170].

### 3.7. Novel Systems

Novel nano-based materials are used in drug delivery systems that discover unique uses of nanostructures, sizes, substances, and phase transitions. The systems listed below are some examples of recent developments in the literature. (1) Silicon-based nanomaterials were developed to deliver anti-cancer drugs selectively with zero-order kinetics to unrespectable tumors. To avoid the discomfort of regular localized treatments, the device can be installed utilizing a less invasive process. Drug delivery methods at the nanoscale that are reusable and biodegradable were also developed [171]. (2) Using electrospinning methods, polymeric nanofibers can be constructed for controlled drug delivery with regulated surface and internal molecular patterns. Polypeptide nanofilms can be applied to other substances’ surfaces or employed as nanodevices with specific structural features [172]. (3) For nano-drug delivery, several physical shapes have been investigated. As promising nano-scale drug delivery platforms, peptide doughnut-shaped nanoreactors have been explored. Depending on the surface features of organic systems, boundary-structured reaction droplets are being used as models to create nanoeggs using inorganic minerals. Systems that do not even depend on the polymeric matrix or lipid shells can still deliver the drugs. A mixture of the PEGylated, protonated analog of the same drug in an aqueous medium can be used to produce hydrophilic drug nanoaggregates simply by introducing an excess of the hydrophilic drugs to the solution [173]. (4) It is possible to modify basic nanoparticles to develop more intricate structures. Particularly, nanoparticles were encapsulated with polymeric nanoshells using layer-by-layer sequential self-assembly. To develop covert or selective nano-scale drug delivery systems, interface alterations can be applied to the nanoshell as a template. To control respiratory drug-release rates after inhalation, nanoparticles were developed as microparticle aggregates containing cleavable in vivo chemical cross-linkages [174]. (5) There is a lot of interest in phase-separated nano-scale drug delivery systems. Amphiphilic copolymers that self-assemble in an aqueous medium can be used to make polymeric nanomicelles, which are highly effective transdermal and oral drug delivery systems for hydrophobic and slightly hydrophilic drugs [175]. Development in the influence of conventional nano-scale drug delivery systems joined by the ability to regulate the structural design, produce novel nanomaterials, and personalize creations, will support engineers, researchers, and medical professionals to exploit nanotechnology for innovative uses in drug delivery.

## 4. Controlling Parameters for DDS

Owing to its unique structures and multi-functionalities, such as mechanical characteristics (compact, better versatility), high selectivity, and flexibility to evaluate the flow path across the human body, drug delivery has attracted great attention as a rapidly developing class of research. It is commonly accepted that their physicochemical characteristics are significantly affected. DDS features (such as drug loading, encapsulation, and coating ligands) could be tuned to enhance therapeutic agent durability, drug release mechanism controllability, and transport efficiency [176].

### 4.1. Drug Loading Strategies

Drug loading strategies must be easy and effective to design an efficient drug delivery system, as this will affect the amount and binding strength of loaded drugs, and it is also crucial to have a good drug-to-system interaction. Interactions that are too strong or too weak will make it difficult to release the drugs or promote unnecessary early leakage, respectively. Likewise, too low drug loading might have an impact on treatment, while too high can have serious impacts, and it is essential to figure out how to make drugs bind to the system. However, covalent bonding, non-covalent adsorption, and direct implantation are the three most common techniques used today to bind different disease-related drugs with the nanosystem [177].

#### 4.1.1. Covalent Bonding

The use of such covalent bonds to bind drugs to the system is a well-known methodology. This approach typically uses ketals/acetals, boronate esters, and Schiff’s base, which are all easily reversible condensation processes. For example, dehydration condensation between –NH_3_ and –COOH was used to immobilize anticancer drugs on the surface of quantum dots. Covalent bonding is regarded as a less versatile method because of the limited number of reversible condensation processes. Furthermore, because of the slower binding and dissociation produced by strong covalent bonds, it takes too long to achieve thermodynamic equilibrium [178].

#### 4.1.2. Non-Covalent Adsorption

Because of its ease of usage and rapid binding and release rate, non-covalent binding has recently become one of the most favored drug loading techniques. Electrostatic interactions, hydrogen bonding, π–π stacking, van der Waals interaction, halogen bonding, coordination bonding, or hydrophilic and hydrophobic characteristics can all be used to adsorb drugs via non-covalent techniques. The halogen bond was recently employed as a hit-to-lead-to-candidate to improve drug-target binding affinity in the context of rational drug design. Some studies looked into anchoring bio-medicines on nanosystems by combining numerous non-covalent interactions, which can provide interaction sites and a stronger affinity [179].

#### 4.1.3. Drug Encapsulation

Encapsulated drugs inside a vesicle generated by a closed phospholipid bilayer membrane is another drug-loading technique. Drug encapsulation, as compared to covalent or non-covalent immobilization, can prevent undesirable early stages of drug tissue interactions. In addition to lipid nano-vesicles, the molecular imprinting technique is being used to directly entrap drugs inside 3D nanomaterials cavities, potentially allowing for molecularly controlled drug delivery. It is used as a molecularly imprinted polymer to entrap the aminoglutethimide substance and develop a drug delivery system. The outcomes of the experiments demonstrate that this material has good biocompatibility and a high drug release rate [180].

## 5. Boron Nitride in Drug Delivery

Nanobiotechnology research has encouraged the development of novel anti-cancer strategies with various types of nanosystems, such as nanosheets, nanospheres nanotubes, nanocarriers, microneedles, nanocapsules, dendrimer-based lipid nanoparticles, nanocarriers, and micelles [181,182,183]. Nanomedicine advances depend on the ability to design and produce distinct nanosystems with appropriate physicochemical properties and biological effects. Among the different nanosystems in biomedical applications, carbon and boron nitride-based materials have grown rapidly in the past few years in the area of drug delivery [184,185,186]. 

Over the past few years, two-dimensional nanosheets have been broadly explored in the area of biomedicine, particularly in oncology therapy [187]. Graphene oxide reformed with biocompatible polymers, such as polyethylene glycol or dextran, has exposed no noticeable toxicity in cellular and animal trials [188]. Moreover, the graphene analogs, such as molybdenum disulfide, tungsten disulfide, and Bismuth selenide nanosheets, have displayed good applications in animal models, but their metabolisms and potential toxicity need to be studied additionally over the period [189]. In recent years, boron nitride nanosheets (BNNSs) have been developed as a new material due to their structural similarity to graphene in the biomedicine area. Moreover, graphene is a zero bandgap semi-metal; however, the BN sheet is an insulator in which the B–N bonds show a partial ionic character [190]. In addition, graphene and BN have covalent functionalization, epoxy, and hydroxyl groups on their surface, which are responsible for their solubility in physiological conditions as well as which enrich their immune biocompatibility and significantly decrease their toxicity [191]. Likewise, many studies have revealed that the cytotoxicity of low-dimensional boron nitrides was as low as that of their carbon analogs. Therefore, BNNSs have great potential to be novel drug delivery systems.

### Anticancer Drug Delivery

Cancer is one of the biggest threats to the public health concerns of the 21st century. The need for the complete eradication of cancer is still untouched, despite several advancements in cancer diagnosis and treatment. The reason for the failure is the severe host toxicity of chemotherapeutic drugs. Confining the bio-distribution of the chemotherapeutic payloads only in the affected cancerous regions can reduce toxicity. Deployment of external stimulus, to release chemotherapeutic agents only in the specified area, is highly beneficial. In Figure 3, we have shown some of the anticancer drug structures.

Attempts to cure cancer employ three principal methods, namely operation, radiotherapy, and chemotherapy. Differing from operation and radiotherapy, which emphasize the treatment of local tissues, chemotherapy is concerned with that of the whole body. While applying chemotherapy, only those groups of cells that are histologically and conclusively diagnosed as a tumor are targeted. Generally, at the stage of diagnosis, the number of the tumor cells already exceeds 1 × 10^9^ (1 g), and the cell population is highly heterogeneous. Therapeutic effects of chemotherapy can no longer be expected in these tumor cells since their growth fraction (GF) is low. To achieve a therapeutic effect, potent therapy is required because the tumor advances to a more intractable state. It is impossible to uniformly kill a group of cells with high heterogeneity because the group of cells is likely to comprise cells that are responsive to the treatment as well as those that are resistant. Thus, it is difficult to obtain a good result through the administration of a single anticancer agent. As a result, therapy combining multiple agents having different mechanisms of action has evolved, i.e., combination chemotherapy. The goal of combination chemotherapy is to eradicate tumor cells through potent therapy before the appearance of resistant cells or an elevation in the number of resistant cells.

Treatment with multiple agents would not have any benefit if the drugs used have antagonistic effects and cancel out the action of each other. Ideally, multiple agents must work synergistically. Thus, it is very important to select drugs according to their phase-specific characteristics to urge more gap phase (G_0_ phase) cells to enter the proliferating cycle to increase the number of tumor cells killed by drugs. Many preclinical studies have shown that drugs such as 5-fluorouracil, 6-mercaptopurine, Gemcitabine, and Capecitabine (Xeloda) are synthesis-phase (S-phase)-selective. Paclitaxel (Taxol), Vinca alkaloids, and Estramustine are mitotic-phase (M-phase)-selective drugs, whereas chlorambucil, Nitrosoureas, Alkyl sulfonates, and temozolomide are non-phase-selective drugs. For high GF tumors such as acute leukemia, phase-specific drugs are firstly used to kill S- or M-phase cells, and then phase non-specific drugs are used to kill tumor cells in other phases, and finally, the above two steps are repeated once again to kill new cells from the G_0_ phase. For low GF tumors such as solid tumors, phase non-specific drugs are firstly used to kill cells of all phases, and then phase-specific drugs are used, and finally, the above steps are repeated to kill the new cells from the G_0_ phases. Such phase-specific anticancer treatment has been explained schematically in Figure 4.

Zhang et al. [70] studied multi-stimuli responsive nanosheets based on hydroxyl boron nitride with palladium nanohybrids (BNNS-OH@Pd). The nanosheets were synthesized by a simple facile thermal substitution method. DOX was loaded into the nanosheets at pH 7.4 through hydrophobic interaction and π–π stacking due to DOX’s aromatic nature. Then, the anticancer drug DOX was loaded onto the system with 32% drug loading efficiency, and the drug release was studied with response to pH, glutathione, and light. The subcellular localization of the nanosheets was examined in the MCF-7 cell line, and cell viability was measured by the MTT assay study. Finally, the therapeutic effect of the nanosheets was studied in in vivo using S180 tumor-bearing mice as the animal model. Feng et al. [72] introduced the folate-conjugated mesoporous silica-functionalized with BN-nanospheres (FA-BNMS). The improved drug loading efficiency was achieved by the incorporation of MS modification. DOX was loaded onto the nanospheres’ BNMS and FA-BNMS complexes with high efficiency (52.6 and 49.2 µg/mg) compared to simple BNNS (6.9 µg/mg). The drug-releasing performance was monitored by various pH (5 and 7.4) conditions, and the maximum drug release was observed at pH 5. A fast DOX release was detected within 8 h (81%) at pH 5, and the steady drug release was observed until 80 h, whereas 26.1% of the drug was released at pH 7.4 after 80 h. FA-BNMS complexes were nontoxic to MCF-7 and HeLa cell lines up to 100 µg/mL concentration, and it was precisely internalized through folate receptor-mediated endocytosis. 

Cheng et al. [73] reported crystalline BNNS-based camptothecin (CPT) delivery. The nanosheets were prepared by a simple NaCl-template reaction with a high yield of 1 g compared to other techniques. For better dispersibility and stability purposes, the nanosheets were hydroxylated (BNNSs-OH). After modifying the stability of the material over two days, there was no precipitation at 1 mg mL^−1^ concentration. The anticancer drug CPT was loaded onto the system with different mass ratios from 1:2 to 2:1, and the drug loading efficiency reached 170 wt%. To evaluate the drug release performance, cytotoxicity of BNNSs-OH@CPT and CPT was studied in 4T1 mouse breast cancer cells, and in vivo experiments were performed in BALB/c female mice bearing 4T1 tumors as the animal model. Permyakova et al. [74] studied folate bonded with BN nanocarriers for targeted drug release. Folic acid was effectively fused with the BNNPs via a simple condensation between the carboxyl group of FA and the amino group of BNNPs. Computational studies also showed that the attachment of FA to the surface of BNNPs does not affect the targeting properties of FA. Sukhorukova et al. [75] described spherical BNNPs (100–200 nm in diameter) with the petal-like surface as a drug release system with respect to pH. The drug loading capacity of the BNNPs was determined by UV–visible absorption and fluorescence, and the calculated drug loading capacity was noted to be 0.055 mg/mg of NPs. DOX-loaded BNNPs were stable at pH 7.4 (neutral), and drug release was observed from the BNNPs at pH 4.5 to 5.5 (acidic). To evaluate the anticancer drug release performance, the cytotoxicity of BNNPs@DOX was studied in IAR-6-1 neoplastic cells. 

Cheng et al. [76] reported dual-stimuli responsive BN nanosheets (BNNSs)-based anticancer drug (DOX) release. A water-soluble adenine-functionalized macromer (A-PPG) self-assembles into the surface of the BNNSs via non-covalent interactions between nanosheets and A-PPG. The nanosheets were easily modified by varying the mass ratio of the BNNS combinations and A-PPG, and the resulting BNNSs showed an excellent response to the pH/temperature. The anticancer drug DOX was loaded onto the BNNSs, and the loading capacity was 36.2%. Under normal physiological conditions, the BNNSs@Dox was quite stable, and the encapsulated drug was released from the nanosheets with respect to increasing the temperature to 40 °C or at pH 5.5. The cytotoxic effects of the nanosheets were performed in RAW 264.7 and MCF-7 cell lines. Feng et al. studied folate-conjugated BNNSs for targeted drug (DOX) release with response to pH. FA was attached to BNNSs by esterification; the drug was loaded onto the system with a capacity of 2.07% and released at pH 5. In vitro cytotoxicity assay exhibited that BNNSs-FA was non-toxic in the HeLa cell line up to 100 µg/mL concentration. Feng et al. [77] introduced pH stimuli-responsive BNNSs fabricated with a charge-reversible polymer for anticancer drug (DOX) delivery. The vapor deposition method technique was used to prepare the BNNSs and then was functionalized with poly(allylamine hydrochloride)−citraconic anhydride (PAH-cit) polymer. The BNNSs-PAH-cit@DOX was prepared via step-by-step electrostatic interaction, and the loading efficiency was around 7.26%. The in vitro cytotoxicity assay was shown in HEK 293, HeLa, and MCF-7 cells, and anticancer activity was studied by the CCK-8 cell line. In addition, high therapeutic efficiency was observed because of the release of the drug into the nucleus of cancer cells. Feng et al. [78] reported cancer cell membrane-based BNNSs for the DOX release with response to pH. The extracted cell membrane (from HeLa cells) was used to encapsulate BNs via physical extrusion. The loading capacity of the BNNSs@DOX was around 86.2%, and the nanospheres released the DOX at acidic pH. Enriched cellular uptake of the nanospheres by HeLa cells was shown due to the homologous targeting of cancer cell membranes. Weng et al. [80] studied water-soluble porous boron nitride material-based drug delivery. For better biocompatibility and effective drug loading, BN materials were hydroxylated, and the drug DOX was released from the system in response to the pH. When the mass ratio of the drug to BN-OH rises from 1:2 to 5:1, the loading capacity also remarkably increases, and the maximum loading capacities were calculated to be 41, 79, and 309 wt%. In neutral pH (7.4), it releases around 36%, while in acidic pH (5.5 to 6.2), it releases around 50–57% of the drug. The biocompatibility and drug transporting properties were studied in in vitro using NIH/3T3 mouse embryonic fibroblast cells and LNCaP cells (human prostate cancer cells). 

Pasquale et al. [69] reported an innovative drug-loaded nanotube with enhanced targeting properties based on the homotypic recognition of glioblastoma cells (GBM- U87 MG cell line). This nanoplatform consisted of boron nitride nanotubes (BNNTs) loaded with doxorubicin and coated with cell membranes extracted from GBM cells. They calculated the drug loading and encapsulation efficiency as 2.15% and 95.6%, respectively. Then, they showed the drug release process at various pH conditions, and the maximum drug release was observed at pH 4.5 at 168 h (0.54 ± 0.04 µg) compared to various pH conditions. They were able to specifically target and kill U87 MG cells without affecting healthy brain cells, upon crossing the blood–brain barrier model. 

## 6. Boronic Acid-Based Drug Delivery

In recent years, many researchers have studied boron-based polymeric nanomaterials for different biomedical applications. Polymeric nanomaterial-based drug delivery systems show great advantages such as prolonged blood flow, ease of modification on the surface, and specific targeting to the site for better efficacy. For better drug accumulation, two main approaches were followed in the DDS: (i) improved affinity between the target site and DDS, and (ii) drug release controlled by some specific triggers such as pH, ROS, or any specific proteins/carbohydrates [192,193,194]. In the 1950s, researchers discovered aryl boronic acid reversibly form cyclic aryl boronic esters in the presence of diols (1,2 or 1,3), later used for saccharide-responsive materials. The diol-binding nature of aryl boronic acid may be modified by the electronic properties of the aryl ring or ligating pendant substituents, and the pKa value of the aryl-derived analogs become significantly lower. This tunability makes aryl boronic acid-based materials useful for saccharide responsive materials. While some of the naturally available biomolecules are used to build the DDS in response to enzymes, it has some disadvantages. In the possible immunogenic response, it shows low stability, and such materials have storage and processing problems. In the case of synthetic chemical-based materials, it has better advantages. These systems have allowed such restrictions but infrequently offer such attractive discernment profiles. While aryl boronic acid-based materials may frequently have some synthetic challenges, they produce a huge component of the efforts towards selectivity, stimuli-responsiveness, and drug delivery research, as this part aims to demonstrate [195,196,197]. The motivation is on insulin delivery systems because of the known covalent reversible chemistry of the aryl boronic acid-based materials for the targeted treatment of diabetes mellitus. In Figure 5, we show a schematic representation of stimuli-responsive boronic acid-based release.

Various advantages of BA materials include biocompatibility, quick reaction times, and high durability. It has been established thus far that pinacol-type boronic ester moieties serve as the most efficient and sensitive monitors for biologically relevant H_2_O_2_ levels. A complex material composition, such as synthetic amplification techniques, can be used to optimize the release rates. The microenvironment’s pH has an impact on the quantitative reactivity between BA and ROS. As a result, using BA as a drug-conjugated linker offers a useful method for managing the release of drugs in response to certain ROS concentrations associated with disease activity. Binding in physiologically important functions is made possible by the special capacity of BAs to link to diol-compounds selectively at physiological pH. There are still many untapped uses for BA polymers, especially for linking saccharides and glycoproteins. The activity of proteins, metabolism and gene regulation, migration, invasion, and other critical activities are all modulated by glycan moieties.

Glycoproteins and carbohydrates are turning into the targets of next-generation therapies because of the crucial function that glycans play in immunological interactions. Employing polymeric nanomaterials results in enhanced selectivity either via enhanced affinity by multivalent heterogeneous receptors or increased affinity using numerous binding sites. To properly exploit these materials, novel BA polymeric materials are necessary that function in neutral to mildly acidic biologically relevant microenvironments. This can occur, for instance, by strategically designing materials or by chemically modifying them to enhance certain qualities, such as hydrophobicity. BA polymeric materials will also be widely used in therapeutic applications owing to novel techniques for regulated synthesis, easy one-step screening, and translatability [198,199].

### Insulin Delivery

Diabetes mellitus, usually mentioned as diabetes, is a chronic disease described by high glucose levels in the blood, which is distinct as a concentration of glucose in the blood >2 g L^−1^ over an extended time. Diabetes is a worldwide disease, which is growing in frequency and is projected to increase to 624 million in the year 2040 [200]. Diabetes combined with other diseases such as cancer, cardiovascular disease, etc., is a most important endangerment to human health and is of rising significance. On the whole, a high level of glucose in diabetic patients is an effect of either lacking insulin production (type 1) or an inefficient response to insulin (type 2) [200,201].

In 1922, Banting and Best identified and isolated insulin [202]. It consists of 51 amino acids organized into polypeptide chains, which are linked by disulfide bridges and form hexameric in the presence of zinc ions [203]. The main task of insulin is regulating carbohydrate metabolism and lipids; on the other hand, it also subdues protein breakdown and influences other functional developments. This means that it can be used for things other than diabetes treatment, such as wound healing, calcium channel blocker poisoning, and anti-aging therapy. Increased insulin secretion is a natural process in response to high blood glucose levels, which triggers cells to absorb and store glucose as glycogen. Cardiovascular disease, nephropathy, ketoacidosis, stroke, nerve damage, and retinopathy are all significant effects of having high-level glucose in the blood for a long time. Diabetes causes an increase in oxidative stress, and the ROS produced as a result contribute to several secondary diabetes problems. Type 1 diabetics are almost solely treated with exogenous insulin, whereas type 2 diabetics may be recommended for small medicines and are urged to adjust their lifestyles. On the other hand, the case of advanced type 2 diabetes might cause an insulin shortage, necessitating the addition of insulin to the therapeutic regimen. This tight regimen must be strictly adhered to to obtain the optimum therapeutic results. Multiple daily dosages, the necessity to match insulin doses to carbohydrate counts at mealtimes, and the timing of insulin delivery at meals all contribute to a high patient burden. Hypoglycemia, which can result in unconsciousness or death, can be caused by high levels of insulin. Moreover, frequent insulin injections and blood glucose self-monitoring can be difficult in social circumstances and can result in physical pain, necrosis, local infections, and nerve damage, limiting patient compliance [204,205,206]. These limitations have drawn the attention of diabetes researchers, who developed a variety of research methodologies ranging from insulin engineering to developing new delivery systems.

Chen et al. [81] reported a boronate-containing glucose-responsive, temperature-stable hydrogel with optimized formulation and assembly into microneedles (MNs) to offer on-demand insulin delivery. The network’s overall hydrophobicity and the degree of intermolecular cross-links were fine-tuned to enable this unique and vital function. The hydrogel had a microporous and linked structure that made it excellent for drug delivery. The two-layer MN patch was created to validate a painless and convenient transdermal distribution of insulin, with the tip area made of hydrogel and the base layer made of crystalline PVA. The prepared MNs patch has sufficient mechanical strength for skin penetration while maintaining the hydrogel’s glucose reactivity. The MNs patch has high biocompatibility and can efficiently penetrate the skin. In comparison to the majority of glucose-responsive MNs patches that rely on glucose oxidase and nanoparticles, synthetic, protein-free, and nanoparticle-free MNs patches could eliminate safety concerns, provide long-term viability, and provide a competitive advantage for large-scale production. Zhang et al. [82] introduced self-regulated insulin release, where a dynamically connected layer-by-layer film made of insulin–polyvinyl alcohol and poly[acrylamide-co-3-(acrylamido)-phenylboronic acid] can be employed. The driving factor for the films was the phenyl boronate ester link between insulin–PVA and P(AAm–AAPBA). When soaked in aqueous solutions, the films progressively break down and release insulin because the link can break under conditions of equilibrium control. The rate of insulin release can be controlled by adjusting the pH and ionic strength. More importantly, it rises as the glucose concentration rises, and it has some advantages for insulin release, including the ability to release insulin over a lengthy period and at a variable rate based on glucose levels. The number of assembly cycles can also be easily altered to change the amount of the drug.

Lee et al. [89] studied trehalose-based hydrogel to stabilize insulin at higher temperatures earlier than glucose-triggered release. The hydrogel was made from a polymer containing trehalose adjacent position and an eight-arm poly(ethylene glycol) (PEG) with phenylboronic acid end-functionalization. The hydroxyls in the trehalose side chains establish boronate ester connections with the PEG boronic acid cross-linker, resulting in hydrogels without any further trehalose polymer modification. The addition of glucose as a stronger binder to boronic acid (Kb = 2.57 vs. 0.48 m^−1^ for trehalose) causes the hydrogel to dissolve, permitting the insulin arrested during gelation to be released in a glucose-triggered manner. Furthermore, after heating to 90 °C, the trehalose hydrogel stabilizes the insulin, as determined by immunobinding. In the presence of the trehalose hydrogel, an enzyme-linked immunosorbent assay detects 74% of insulin after 30 min of heating, but just 2% was found without any additives. Cai et al. [86] reported that hydrogel-based glycopolymer was synthesized by RAFT polymerization via boronic ester linkages. At physiological pH, the glycopolymer hydrogels were made by phenyl boronate-diol cross-linked binding and showed glucose-sensitive performance. The hydrogel was synthesized via a dynamic covalent link by the reaction of boronic acid with diols. After incubation with an aqueous solution, a hydrogel with a typical porous structure showed a quick increase in swelling equilibrium, up to 1856%. The hydrogel demonstrated enhanced drug loading capabilities of up to 15.6% using insulin as a model protein therapy, as well as glucose-responsive insulin release under a physiological environment. In addition, the viability of NIH3T3 cells after treatment with hydrogel was greater than 90%, demonstrating that the hydrogel was non-cytotoxic.

Peng et al. [87] described a bio-hydrogel-based composite with a response to pH/glucose that was synthesized at room temperature by an electron beam irradiation method. The appearance of carbonyl in the polymerization of 4-ethenyl-phenylboronic acid, grafting and cross-linking processes in composites, and the formation of a novel composite hydrogel between poly-4-ethenyl-phenylboronic acid and the cellulose matrix were all caused by electron beam irradiation. Composite hydrogels with pH and glucose-sensitive properties were made by including phenylboronic acid groups, and glucose-responsive features were explored by the self-regulation of insulin release of composite hydrogel over a serial glucose solution with various concentrations. The suggested composite hydrogel’s cleverness and biocompatibility make it a suitable candidate for a variety of applications, including self-regulated drug delivery and actuators, devices, and glyco-sensitivity in the separation process. Zhi et al. [88] reported nanofilaments bioconjugates with diol affinity-based single hydrogel with stimuli-responsiveness, injectability, self-healing, and customizable internal architectures. The M13 virus with a high aspect ratio was used to create a hydrogel by linking a tailor-made low-pKa phenyl boronic acid analog to a well-defined green nanofiber (PBAM13). Multiple diol-containing substances, such as poly(vinyl alcohol), were used to crosslink PBA-M13 via the conventional boronic diol dynamic bonds, resulting in dynamic hydrogels with adjustable mechanical strength. The produced hydrogels had good injectability and self-healing properties, as well as chemical access to the PBA moieties on the virus backbone within the gel matrix. Simple shear-induced alignment of viral nanofibers was used to impart ordered internal structures to virus-based hydrogels. Moreover, in situ gelation generated by diffusion of diol-containing molecules was used to fix the chiral liquid crystal phase of the PBA-M13 virus, resulting in unique hydrogels with chiral internal structures. The sugar sensitivity of this gel enables payloads such as insulin to release in a glucose-regulated manner. At physiological pH, all of these qualities have been accomplished.

Tong et al. [93] studied glucose-responsive hydrogels for insulin release. The dynamic boronic esters linkages between phenylboronic acid-loaded γ-polyglutamic acid (PBA-PGA) and konjac glucomannan were developed in situ using a glucose-responsive hydrogel (KGM). It was expected that using the hydrogel as a delivery vehicle for Ins/Lir would improve the latter’s cumulative influence on suppressing DN progression. The hydrogel demonstrated excellent glucose-responsive abilities in scan electronic microscopy and rheological experiments. Under hyperglycemic conditions, the glucose-dependent release rate between either Ins or Lir from hydrogel was consistently observed. On streptozotocin-induced diabetic rats, the preventative efficacy of a hydrogel encapsulating insulin and liraglutide (Ins/Lir-H) on DN progression was also investigated (DM). The morphology of the kidneys was recovered six weeks following a single dose of Ins/Lir-H, as evidenced by ultrasonography, and the renal hemodynamics was significantly enhanced. Moreover, urine protein and albumin/creatinine levels were well controlled after 24 h. Oxidation and stiffness were also significantly reduced. In addition, after therapy with Ins/Lir-H, renal NPHS-2 was significantly enhanced. Ins/Lir-therapeutic H’s mechanism was found to be strongly linked to the inhibition of lipid peroxidation and the stimulation of phagocytosis. Guo et al. [94] described sugar-responsive nanogels for insulin delivery. 3-acrylamidophenylboronic acid as a glucose identifying component, 2-(acrylamido)glucopyranose as a biocompatible functional group and boron dipyrromethene as a fluorescence nucleophile were covalently integrated into sugar responding nanogels. Reversible addition-fragmentation chain transfer (RAFT) polymerization was used to make the nanogels in a water/ethanol mixture. After being treated with 3mg/mL glucose medium, nanogels were able to respond to glucose, and their size expanded. The intensity of fluorescence nanogels varies considerably depending on the glucose content. Furthermore, insulin can be incorporated into the nanogels with a loading of up to 8.2%. The presence of phenylboronic acid components in nanogels and glucose levels in the release media impacted the drug release process. The cytotoxic effect of nanogels was evaluated, and it was observed that nanogels were biocompatible.

Zhao et al. [91] introduced dual-responsive injectable hydrogels for insulin delivery. Using phenylboronic adapted chitosan, poly(vinyl alcohol), and benzaldehyde encapsulated poly-ethylene glycol, dual responsive pH and glucose inject hydrogels were developed by cross-coupling Schiff’s base and phenyl boronate ester. During in situ crosslinking, protein drugs and live cells may be integrated into the hydrogels, resulting in prolonged and pH/glucose-stimulated drug release from the hydrogels, as well as biocompatibility and propagation in the three-dimensional hydrogel network. Therefore, the hydrogels containing insulin and hepatocytes were recognized as bioactive wound dressings for the diabetic healing process. The efficiency of hydrogel dressings in tissue repair was tested using a streptozotocin-induced diabetic rat model. The results showed that integrating insulin and L929 into hydrogels could enhance neovascularization and collagen deposition, as well as improve diabetic wound healing. Elshaarani et al. [92] reported hydrogels with chitosan reinforcement for glucose sensing and insulin delivery. Using poly(ethylene glycol) diacrylate as a crosslinker, unique hydrogels containing phenylboronic acid-co-chitosan-loaded maleic acid were developed. Because of the 2:1 boronate glucose binding, the synthesized hydrogels were constricted at low glucose levels and expanded at high glucose levels due to 1:1 boronate glucose complex formation. For glucose sensing and insulin delivery, both binding mechanisms play an important role. The incorporation of CSMA into the hydrogels matrix improved not only the sensitivity to glucose at physiological pH, but also the mechanical capabilities and encapsulation efficiency of the hydrogels. 

Chen et al. [83] described an enzyme-free polymeric materials-based microneedle-array patch for insulin delivery. The MN-array patch developed from our smart gel might be used as an on-skin patch. Mechanically resistant silk fibroin (SF) obtained from the silkworm Bombyx mori was incorporated into the gel network to maintain the necessary stiffness for skin penetration. SF has good biocompatibility, tensile stability, and a programmable degradation rate to control the rate of crystalline-sheet areas via post-processing. An MN-array patch was constructed by combining SF with a hydrogel semi-interpenetrating network. In an aqueous environment, our composite MN-array patch remains stable over 2 months and exhibits both persistent and acute glucose-responsive insulin release abilities. Chen et al. [84] studied smart microneedles for glucose-responsive insulin delivery crafted from silk fibroin-mixed semi-interpenetrating network hydrogel. The insulin delivery system is made from hydrogel, biocompatible silk fibroin (SF), and phenylboronic acid/acrylamide. Six fabrication strategies were studied to keep the hydrogel’s glucose sensitivity while minimizing distortion during fabrication. The preferred route for developing efficient MNs was to obtain a two-layer approach, with a needle region composed of SF-linked hydrogel and a base layer made of SF. Through the control of the skin layer developed on the surface, the hybrid MN released insulin spontaneously in response to the glucose change pattern. Moreover, after 1 week in an aqueous environment, this hybrid MN maintained its unique needle shape, minimizing the safety issues associated with dissolving MNs and demonstrating the feasibility of sustained delivery. Yuan et al. [95] introduced multi-responsive nanogels for insulin delivery. N, Ndiethylacrylamide, and 4-vinylphenylboronic acid were used as monomers in the emulsion precipitation polymerization of multi-responsive nanogels that are sensitive to pH, temperature, and glucose levels. The effect of the monomer feeding ratio and emulsifier amount on the size and size distribution of the nanogel was evaluated. The width of the nanogel was 186.9 nm under optimal conditions, with outstanding monodispersity (PDI 0.005), encapsulation efficiency of 88.67%, and insulin loading efficiency of 17.73%. The glucose concentration might be used to modulate and control the release rate of insulin encapsulated in nanogels, allowing for intelligent modification and control of blood glucose levels in a steady range of concentration. The insulin-loaded nanogels steady release of insulin in vitro can be rationally controlled by the glucose concentration, indicating a potential application in the field of self-regulated drug delivery systems. 

Gu et al. reported [90] microgel-based thermo- and glucose-sensitivity and enhanced salt tolerance for regulated insulin release in a physiological environment. A precipitation emulsion technique was used to make such an effective microgel from a thermoresponsive (N-isopropyl acrylamide), glucose-sensitive ((2-phenylboronic esters-1,3-dioxane-5-ethyl)methyl acrylate), and water-soluble crosslinker (poly(ethylene glycol) diacrylate). Upon adjustments in temperature and ionic strength, these colloidal nanoparticles displayed dependent responsive behavior. Among them, the microgel with 20.7 mol % and a narrow particle size distribution is appropriate for diabetes treatment because it can adapt to the various glucose levels in the extracellular environment over a clinically significant range (0–2.0 mgmL^−1^), controlled release of loaded insulin, and is extremely stable under physiological circumstances. The insulin release from the microgels seemed significantly reliant on glucose levels, according to the in vitro release studies.

Kataoka et al. [207] studied phenyl boronate ester linkage-based polyplex micelles (PMs) for mRNA delivery with response to ATP concentration. By complexing mRNA with poly(ethylene glycol)-polycation block copolymers derivatized with phenylboronic acid and polyol groups, which form crosslinking frameworks via spontaneous phenyl boronate ester formation, we created mRNA-loaded polyplex micelles (PMs) with ATP-responsive crosslinking in the inner core. Thusly produced PMs are resistant to enzymatic attack and, when stimulated by the breakage of phenyl boronate ester links in response to increased ATP concentration, disintegrate in the cytosol to release mRNA. Owing to the changing geometry between the robustness in the physiological environment and the ATP-responsive mRNA release in the cytosol, two significant components of the PM, including (i) the introduction ratios of phenyl boronate ester crosslinkers and (ii) the framework and protonation degree of substituents in the polycation section, are crucial for enhancing protein expression in cell cultures. In order to prolong blood circulation following intravenous injection, cholesterol moieties were added to the mRNA and ω-end of the block copolymer, substantially stabilizing the ideal PM formulation. In comparison to the control preparation with improper synthetic modulation, the ideal mRNA PM formulation was created via required chemical modulation, where it meets high intracellular translational interaction and stabilizes the mRNA system in the challenging in vivo environment of the bloodstream. In vivo use appears to be the ultimate objective of PM systems for mRNA delivery. Kataoka et al. [208] described phenyl boronate ester linkage-based RNA oligonucleotide compounds with polyplex micelles for effective mRNA delivery. A substantial but reproducible connection between mRNA and polycation is necessary for polyplex for messenger RNA (mRNA) delivery in order to achieve external resilience and specific intracellular breakdown. Here, mRNA and polycation-bridging RNA oligonucleotide (OligoRNA) compounds are created to stabilize polyplex micelles (PMs). A group of OligoRNAs that have a polyol moiety added to their 5′ ends is intended to hybridize to specific locations along the mRNA strand. Phenylboronic acid (PBA) substituents create reversible phenyl boronate ester bonds with polyol substituents at the 5′ ends of OligoRNAs and a diol molecule at their 3′ end ribose, in the PM core, following PM preparation from hybridized mRNA and poly(ethylene glycol)-polycation block copolymer derived with PBA conjugates in its cationic section. The OligoRNAs serve as a connection to link ionically complexed mRNA to polycation, enhancing PM durability in the extracellular medium and protecting it from ribonuclease attack and polyion exchange reaction. During cellular absorption, increasing intracellular adenosine triphosphate levels causes the breakage of phenyl boronate ester bonds, leading to the release of mRNA from the PM. The PM essentially demonstrates the capabilities of the bridging technique in polyplex-based mRNA delivery by efficiently introducing mRNA into cell lines and mouse lungs upon intratracheal delivery. 

Kataoka et al. [209] introduced phenyl boronic acid-based polyplex micelles for plasmid DNA (pDNA) delivery. Using block catiomers derived from poly(ethylene glycol) and derivatized with 4-carboxy-3-fluorophenylboronic acid (FPBA) and D-gluconamide to construct pH- and ATP-responsive linkages in the core, we synthesized plasmid DNA (pDNA)-loaded polyplex micelles (PMs). These PMs demonstrated robustness in the extracellular environment and smooth endosomal escape following cell viability, and they aided in the decondensation of pDNA brought on by a rise in ATP levels within the cell. Laser confocal microscopic analysis showed that FPBA construction increased the endosomal escapability of the PMs; this effect most likely came about because the release of block catiomers with hydrophobic FPBA conjugates in the chain length from the PM at lower pH conditions of endo/lysosomes assisted endo/lysosomal cell damage. Additionally, the profile of intracellular pDNA decondensation from the PMs has been observed using flow cytometry and Förster resonance energy transfer measurement. These findings demonstrated that PMs designed for ATP responsiveness effectively decondensed loaded pDNA intracellularly to achieve promoted gene transfection. Kataoka et al. [210] reported an ATP-triggered phenyl boronate-based micelle for siRNA release. Creating polyion complex (PIC) micelles, which naturally form in aquatic media due to electrostatic interactions between the anionic siRNA and cationic polymers, is one possible strategy. Moreover, in general, these PIC-based transporters exhibit instability in physiological settings. This instability is principally caused by the siRNA’s 20–25 nucleotide chain length, which is quite short and leads to poor thermodynamic stability. Hence, it has been interesting to stabilize the PIC-based carriers to allow for planned breakdown once they reach the intracellular destinations (to release siRNA). Modern research has concentrated on one or a combination of the three representative techniques listed below: covalent conjugation of siRNAs to a homing polymer, the addition of hydrophobic materials to strengthen the core aggregation, and disulfide bonding to crosslink the core aggregate. As a result, combining such strategies frequently produces a highly complex system and preparation strategy. Our findings suggest that an ATP-triggered siRNA release from an intracellular PIC micelle can be achieved. Further fine-tuning is made possible by the concentration levels that can be regulated to maintain complicated stability. On the basis of the functional groups, it is also possible to regulate the strength of the ribose–PBA interaction as well as the hydrophobicity of PBA, both of which are key considerations of complex stability. Enhanced biocompatibility and endosomal escape might be achieved by altering the kind and length of the opposite polycations. In addition, with in vitro and in vivo studies of gene silencing, more initiatives are being made in the direction of these capabilities.

Smith et al. [211] described anionic saccharides that trigger liposomes carrying phospholipids containing boronic acid for calcium ion fusion. Technical problems with extended liposome lives and liposome targeting have subsequently been resolved, but the issue of ineffective cell transfection is still up for discussion. While many techniques are known to cause liposome release, there are relatively few techniques for causing fusion and content mixing. The liposome method described in this article is the first to be stimulated for calcium ion fusion by nontoxic anionic saccharides. This opens the way to the development of “sugar-sensitive liposomes”, or liposomes that can be selectively induced to undertake cell fusion and transfected by a relatively high dosage of anionic saccharide. Our findings and those of others hint at the possibility that utilizing molecular recognition to promote the creation of noncovalent oligomeric regions at the membrane point of contact is a useful strategy for designing a membrane-fusing technology from a supramolecular chemistry approach. Smith et al. [212] utilized liposomes with a synthetic carbohydrate-binding domain to increase cell binding. Comparing appropriate control liposomes with DOPEB, a structurally identical phospholipid that contains a boronic acid residue, to related control liposomes carrying DOPEBA, we discovered that the latter exhibit increased attraction for erythrocyte ghosts (red blood cells). The easiest and most well-studied human cells are erythrocytes. Because they are unable to go through endocytosis, they make excellent analogs for binding and fusion experiments. Using fluorescently tagged liposomes having 0.3% of the phospholipid probe and its resonance energy transfer quencher Rh-PE (N-(lissamine rhodamine B sulfonyl)phosphatidylethanolamine), liposome–cell interaction and membrane fusion have been observed. We observe that liposomes containing the water-soluble pigment sulforhodamine B were used in fluorescence imaging investigations, but no observable indication of improved transport of liposomal contents was found. Utilizing existing supramolecular capabilities, the generation of surface-functionalized liposomes that bind to certain target cells looks to be possible. The creation of structures that cause membrane fusion and thus enhance drug delivery is a more challenging task.

Best et al. [213] studied carbohydrate binding-driven boronic acid liposomes for cellular delivery and content release. We have discovered that boronic acid lipid 1-containing liposomes are efficient for content release and carbohydrate-driven cell penetration. These liposomes might offer flexibility either by releasing content just outside of the cells or by effectively delivering therapeutic payload through the cell entrance, both of which are expected to improve delivery. This offers a promising method for general cellular targeted delivery, but it also makes it possible to target sick cells specifically for distribution based on the unique makeup and abundance of cell surface carbohydrates. Best et al. [214] reported bis-boronic acid-based liposomes for carbohydrate detection and cellular delivery. In order to obtain sensitive saccharide sensing and improve cell surface recognition based on carbohydrate-strong interactions, we describe a liposome platform containing bis-boronic acid lipids (BBALs) to enhance valency. Different BBALs (1a-d) with varied linkers between the binding components were developed and synthesized in order to alter their characteristics. According to a microplate fluorescence-based analysis of carbohydrate binding, these molecules have different binding characteristics depending on their structural makeup. Fluorescence microscopy tests further showed that the incorporation of BBALs into liposomes improved cellular interaction. These findings show that multivalent BBALs are a novel glycan-attaching liposome method for targeted delivery.

Kataoka et al. [215] studied phenylboronic ester-based polymeric nanoreactors (NRs) for effective in vivo anticancer treatment, including tumor-specific activation and self-destruction. Using a diblock copolymer composed of poly(ethylene glycol) (PEG) and copolymerized phenylboronic ester or piperidine-functionalized methacrylate (P(PBEM-co-PEM)), unique glucose oxidase (GOD)-loaded therapeutic vesicular NRs were developed. The tumor pH (pH 6.5–6.7) and H_2_O_2_ responsive hydrophobic regions of NRs are represented by the PPEM and PPBEM portions, respectively. NRs remain in an inactive state in healthy cells with a pH of 7.4. Owing to protonation at tumor acidity, PPEM portions become hydrophilic after building up in tumor tissues, increasing the susceptibility of NRs membranes and enabling the transfer of nutritional small molecule compounds (glucose and oxygen). The oxidation reaction is catalyzed to produce enormous amounts of H_2_O_2_ by the presence of GOD. The chronic stress in the tumor site rises in tandem with the decrease in nutritional material content. In contrast, the high H_2_O_2_ level damages PPBEM segments and causes the vesicles to self-destruct, releasing quinone methide (QM) as a residue. The capacity of QM to reduce intracellular GSH affects the capacity of cancer cells to resist oxidative stress. Raising oxidative stress and decreasing GSH work together to effectively destroy cancer cells and stop the expansion of tumors.

## 7. BODIPY-Based Drug Delivery

Boron-dipyrromethene and its derivatives are extensively used fluorescent dyes in a variety of chemical and biological applications. For S0-S1 transitions, BODIPY normally shows a strong wide absorbance peak within the visible range, as well as a shoulder band owing to the 0–1 vibrational transition. Its constrictive planar structure, combined with its high molar absorptivity and high quantum yield, photocatalysis and thermal stability, switchable spectral properties, and ease of structural modification, make it and its derivatives highly fluorescent dyes with a wide range of chemical and biological applications, which include fluorescent sensor systems [216], triplet photosensitizers [217], drug delivery [218], photodynamic therapy (PDT) [219], photocatalytic hydrogen generation [220], bioimaging [221], organic light-emitting devices [222], dye-sensitized solar cells [223], triplet-triplet annihilation upconversion [224] and commercial biological labels [221]. A few review articles have been published that summarize the design, synthesis, and photophysical nature of BODIPY, along with polarized luminescence spectroscopy, and a summary of biomedical applications such as near-infrared fluorescent probe, pH and metal ion indicators, and the detection of H_2_S, NO, CO, bio-thiols, and reactive oxygen or nitrogen species in living organisms [225]. Figure 6 shows the caging at various positions of the BODIPY fluorescent probe and stimuli-responsive drug release. This section focuses on the various stimuli-responsive anticancer drug release utilizing the BODIPY chromophore.

Zhang et al. [97] described the fluorinated aza-BODIPY (**1**)-based nanoplatform for anticancer drug (DOX) release (Figure 7). By inserting nonadecafluorodecanoic acid into aza-BODIPY via the amide linkage, fluorinated azaboron-dipyrromethene with excellent near-infrared absorption is synthesized. Nanoparticles are synthesized utilizing co-precipitation techniques, and the resulting NPs can not only load with DOX with a high loading efficiency (25%) but can also absorb PFC droplets (1H-perfluoropentane) with a bp of 42 °C due to the fluorinated links inside the NPs. The hyperthermia influence of NBF helped promote the liquid–gas phase transformation of PFC droplets upon 808 nm light irradiation, triggering the controlled release of DOX and boosting echo signals for ultrasound imaging. Considerable progress in preventing tumor progression is accomplished with the NPs under light irradiation, as demonstrated by in vivo ultrasound imaging and photoacoustic imaging, with no observable adverse effects. The NPs incorporate excellent bioavailability, stimuli-responsive drug delivery, multi-mode imaging, cancer anoxia relief, and combined PTT/chemotherapy, also expanding the use of aza-BODIPY in cancer treatment.

Li et al. [98] studied BODIPY-grafted water-soluble chitosan-based nanoshells for drug release (**2**). The loaded drug methotrexate into the nanoshells resulted in chemotherapy, photothermal treatment, photodynamic therapy, and imaging in one platform for combinatorial treatment in HeLa cells (Figure 7). The nanoshells can act as drug transporters, increasing solubility, decreasing toxicity, and improving the utility of combinatorial PTT and PDT. The reduced dosage of chitosan-based nanoparticles (IC_50_ = 34.5 g/mL) is lower than the corresponding drug MTX (IC_50_ = 56.9 g/mL), indicating good biocompatibility and safety of chitosan-based BODIPY nanoshells as carrier materials. Photothermal temperature and MTT tests demonstrate that NPs generate ROS and have high photothermal conversion efficiency (38.3%) when exposed to light. Meng et al. [99] introduced pillar-layered metal–organic frameworks based on BODIPY for chemotherapy and PDT (**3**). A unique two-fold interpenetration pillar-layered MOF material was successfully synthesized employing the BODIPY-derivate bipyridine as a linker and photosensitizer, to act as a drug carrier and photodynamic agent at the same time (Figure 8). Upon 660 nm light irradiation, the system could produce singlet oxygen, while the drug doxorubicin could be loaded for chemotherapy. The system had a high drug loading efficiency of 49.7%, regulated drug release, and excellent biocompatibility. In Hela cells, the cellular uptake was evaluated at 1, 4, and 8 h, and high amounts of ^1^O_2_ were produced during light irradiation. 

Song et al. [101] reported a near-infrared Azo-BODIPY (**4**)-based cancer-targeting prodrug Biotin–Gefitinib with real-time fluorescence visualization that releases the drug with respect to GSH concentration (Figure 8). Gefitinib, an effective anticancer drug, combines a biotin-recognizing ligand with a GSH-responsive disulfide bond linker to produce the prodrug Biotin–Gefitinib. We synthesized fluorescent theranostics by attaching a BODIPY probe to the molecular structure of the prodrug Biotin–Gefitinib. The high level of GSH in the pathological environment allows the prodrug Biotin–Gefitinib to enable Gefitinib release. We utilized the fluorescent system to evaluate the drug delivery of the prodrug Biotin–Gefitinib in PC9 cancer-bearing nude mice models, suggesting that it can be used to track in vivo drug release activity. In comparison to single Gefitinib chemotherapy in cells and in vivo, the prodrug Biotin–Gefitinib can be tailored to accumulate inside the cancer site, providing a higher and much more effective therapeutic response. The fluorescent images also demonstrate that anticancer drug-targeted growth and continuous retention within tumor sites lead to improved therapeutic efficacy. Wang et al. [102] introduced azobenzene-based BODIPY (**5**) fluorophore with azoreductase-responsive drug delivery for treating hypoxic tumors and colon disease (Figure 9). Different azobenzene compounds have attracted interest in developing fluorescent dyes, even though most azobenzene compounds exhibit absorptions and emissions that are in the visible region only. The NIR fluorescent probe BODIPY was attached with an azobenzene group for quenching the fluorescence. The azo bonds were cleaved in the presence of azoreductase, and the probe’s fluorescence switched from “OFF” to “ON”. Moreover, the drug-loaded nanoprobe was successfully synthesized by self-assembly of the amphiphilic polymers, which were generated from PBS buffer solution. In a simulated colon environment, these drug-loaded micelles show an effective switch of NIR fluorescence associated with the release of the drug under the action of azoreductase. The MTT assay was used to assess the cell cytotoxicity of the system toward the L929 normal cell line and CT26 cancer cell line. The micelle solutions against the L929 cell line showed almost no cytotoxic effects at all tested concentrations (cell viability >99%), while the blank micelle solutions against the CT26 cell line showed very low toxicity (cell viability >95%) after 72 h of incubation with the L929 and CT26 cell lines, respectively. The polymeric nanoparticles exhibit excellent biocompatibility for prospective uses in biological detection and controlled drug delivery.

Porubsky et al. [103] studied amino-BODIPY (6,7)-based targeted drug release with real-time monitoring (Figure 10). Two-drug delivery approaches are based on small-molecule drug conjugates with selective targeting and drug release tracking by ratiometric fluorescence. The functionality of these approaches was confirmed by three model systems: (i) amino-BODIPY for the detection of the cleavage process in real time, (ii) angiogenesis in the solid tumor was targeted by peptide-specific receptors or FRET cleavage monitoring using the red-BODIPY moiety, and (iii) a quinolinone-based model drug. Model drug release is based on a self-immolating disulfide linker that is sensitive to thiol-rich environments, particularly glutathione, which is overexpressed in tumor cells. The findings suggest thiol-mediated cleavage of the fluorescent reporter and specific drug release in a tube. By interacting with integrin receptors, the compound was proven to permeate the cells.

Lu et al. [104] reported a PEGylated dimeric BODIPY-based nanocarrier (**8**) for combination therapy (Figure 11). The production of ROS in the product was greatly reduced by conjugating a cathepsin B substrate peptide into the system, owing to the restriction of the electron-donating amino group. After the peptide linkage was cleaved by cathepsin B, in vitro experiments showed that ROS production was resumed under laser irradiation. The system was subsequently PEGylated and conjugated with a peptide to encapsulate 10-hydroxycamptothecin (hydrophobic anticancer drug). The synthesized nanoparticles had good stability in serum-containing solution with a size of ~200 nm. The combination of cathepsin B-activated PDT and chemotherapy was effective in suppressing the development of 4T1 breast cancer cells while also inducing cell damage. The nanoparticles were also capable of penetrating the 4T1 3D tumor spheroid and efficiently reducing the size.

Asem et al. [105] described the surface engineering of polymeric nanoparticle post-polymerization-induced self-assembly-based nanocarriers for drug delivery. Using RAFT-driven emulsion polymerization via polymerization prompted self-assembly; we synthesized functional poly(acrylic acid)-b-poly(butyl acrylate) NPs. The chain extended with the hydrophobic monomer n-butyl acrylate, yielding stable, homogeneous, and repeatable NPs with a diameter of ~130 nm. A two-step strategy was used to investigate the surface engineering of the NPs. Under moderate circumstances, the hydrophilic NP-shell was changed with allyl groups using allyl amine, resulting in stable allyl-functional NPs appropriate for bioconjugation. The allyl-NPs were conjugated with a fluorophore (BODIPY-SH) to the allyl groups utilizing thiol-ene click chemistry. UV–vis spectroscopy confirmed the effective attachment of BODIPY-SH to the allyl-NPs, revealing the typical absorbance of the fluorophore at 500 nm, and DLS revealed that the NPs sustained both emulsion stability and monodispersity. The NPs and allyl-NPs did not cause any cytotoxicity in the RAW264.7 or MCF-7 cells, suggesting that they have low toxicity. The in vitro cellular uptake of NPs by the J774A cell line was evaluated to be time and concentration sensitive. The drug doxorubicin was incorporated into the NPs with high efficiency of 90%. A regulated release pattern was also seen throughout 7 days after a tiny initial rapid release within the first 2 h. Shen et al. [106] reported a BODIPY-based amphiphilic polymer for photothermal enhanced anticancer drug Docetaxel release with response to laser irradiation (**9**). The BODIPY tail, a hydrophobic polylactide segment, and a hydrophilic polyethylene glycol section were synthesized effectively (Figure 11). The BODIPY could have a strong absorbance within the NIR region and serve as a photothermal agent because of its narrow bandgap energy. The therapeutic drug docetaxel (DTX) can be enclosed inside the core of self-assembled nanostructures owing to hydrophobic interactions between the drug and hydrophobic regions. The synthesized nanoparticles enhance drug release in an aqueous medium in response to heat. Utilizing the photothermal effect as well as the improved penetration and retention effect, the customized release of drugs and high tumor penetration of nanomaterials can be accomplished in the lung cancer A549 cell line, leading to improved therapeutic efficiency and minimizing undesired side effects. In vivo anti-tumor studies also show that photothermal-enhanced chemotherapy efficiently reduces tumor growth, whereas systemic toxicity and side effects of DTX are reduced significantly owing to functionalization. Long et al. [107] introduced a simple method for fabricating light-responsive nano assemblies by coassembling the BODIPY-chlorambucil prodrug (**10**) and the NIR dye IR783 to obtain optimum prodrug loading capacity (99%) and effective photoresponsive prodrug activation (Figure 11). The IR783 dye not only stabilized the NPs and enhanced tumor targeting as expected, but it also degraded upon light irradiation and could be used to monitor NPs disintegration in real time utilizing fluorescent imaging. The NP dissociation, anticancer drug release, and singlet oxygen generation were observed when exposed to 530 nm green light, demonstrating a photoresponsive antitumor activity. Upon light irradiation, the NPs showed an “ON-to-OFF” pattern, allowing for in-situ fluorescence imaging of the photoresponsive disintegration of the NPs. IR783’s sulfonate groups also allowed for increased tumor accumulation of the NPs via CAV-1-mediated transcytosis. The photoresponsive anticancer efficacy of the NPs that included tumor-targeting, fluorescence monitoring, and photoresponsive therapeutic activity was confirmed in both in vitro and in vivo experiments, leading to more efficient tumor eradication in HCT116 tumor-bearing mice. 

Porubsky et al. [108] described amino-BODIPY (**11**–**13**)-based glutathione cleavable conjugates for anticancer drug release (Figure 11). With the utilization of a self-immolating linker in the conjugate synthesis, free amino-BODIPY was released along with the drug. Asymmetrical linker was utilized to connect an anticancer drug with such an amino or hydroxyl group, although an asymmetrical linker was used to link an anticancer drug with its thiol group (Figure 1). In the presence of thiols, the disulfide bridge acts as a switch, stimulating the release of the anticancer drug with unbound amino-BODIPY. The rate of drug delivery is controlled by both the GSH level and the solution of the pH. The OFF–ON function and ratiometric fluorescence monitoring of the anticancer drug release were further verified in the HeLa cell line using native and significantly raised GSH concentrations. Moreover, UV–vis spectrometry enables the calculation of conjugate concentrations irrespective of cleavage extent. The approach has also been utilized for making a molecular electronic selector. Two logic gates (AND and NAND) controlled this selector, which allowed for the irreversible changing of the two different sources (GSH and pH) while maintaining the intensity of one light. Chang et al. reported [109] pH-responsive BODIPY (**14**)-based polymeric nanocarrier for anticancer drug DOX release and photodynamic therapy (Figure 11). The polymeric platform was synthesized from the photosensitizer core diBr-BODIPY and MPEG as the hydrophilic side chain. The anticancer drug DOX was effectively encapsulated into the self-assembled nanosystem with a loading capacity of 6.73%, and the resulting drug-loaded nanosystem remained stable in physiological circumstances but showed efficient and faster drug release in an acidic medium due to the cleavage of the acid-sensitive hydrazone bond linker. The drug-loaded nanosystem was found to have high biocompatibility for imaging-guided chemotherapy and PDT as well as a high level of effectiveness and safety. The significant intracellular fluorescence of the internalized drug-loaded nanosystem in the HeLa and MCF-7 cell lines was demonstrated by confocal laser scanning microscopy. In addition, the nanosystem generates singlet oxygen upon light irradiation. Long et al. [112] studied red-light-responsive trigonal molecule BODIPY (**15**)-based nanoparticles for anticancer drug paclitaxel release. The self-assembled NPs were synthesized from the trigonal molecule, BODIPY, and bounded with platinum (II) tetraphenyl-tetra benzoporphyrin (Figure 11). The anticancer drug was loaded into the self-assembled NPs by the flash nanoprecipitation technique. During the self-assembly method, the drug-loading capacity of the NPs grew from 5.6 to 13.1%, while the weight ratio of the NPs increased from 5 to 80%. The NPs disintegrated and exhibited efficient drug release performance after being irradiated with red light (635 nm), leading to enhanced anti-tumor activity in vitro and in vivo. Utilizing confocal laser scanning microscopy, light-triggered drug release and subsequent cellular uptake were studied in the murine breast cancer 4T1 cell line. For in vivo therapeutic efficiency, a breast cancer model was constructed utilizing 4T1 tumor-bearing mice. Sozmen et al. [113] reported light-triggered BODIPY and chitosan-based nanoparticles for anticancer drug DOX release and photodynamic therapy (Figure 2). The organic NPs were synthesized through the ionic interaction between BODIPY and chitosan hydrochloride at pH 6. The anticancer drug DOX was loaded into the chitosan NPs, and the pH of the solution was maintained at ~6 (color changed from pink to purple). However, when both drug-free (NPs) and DOX-loaded organic NPs exhibited excellent PDT activities, they were found to be more efficient on the MCF7 cancer cell line and less cytotoxic on L929 cells.

## 8. Boron-Based Materials for Other Diseases

In recent years, boron-based materials have been used for different diseases such as Alzheimer’s disease (AD), Parkinson’s disease, antimalarial, antimicrobial, anti-inflammatory prodrugs, COVID-19, etc. 

Liu et al. [226] reported dendrimer-peptide conjugate-based ROS-responsive delivery to target the AD microenvironment and decrease the inflammatory responses at an early stage. The nanosystem was synthesized from three components: peptide, ROS-responsive ethylene glycol-based boronic dendrimer and clearing capability, and nuclear factor (erythroid-derived 2)-like 2 (Nf2)-based therapeutic peptide. The synthesized nanosystem was able to cross the blood–brain barrier (BBB) and bind to the glycation end-product RAGE and is widely expressed in the microenvironment of Alzheimer’s disease. It had a synergistic impact of enhancing antioxidative efficiency and decreasing glial cell reactivity by removing ROS and releasing Nf2. In vitro and in vivo studies have shown that inhibiting inflammatory responses has neuroprotective effects in the early stages of Alzheimer’s disease. However, it shows that multi-target therapy can improve therapeutic results in the early stages of Alzheimer’s disease when compared to single therapy and that it may have a greater clinical translation potential. Maiti et al. [227] described boronic acid-based materials for the suppression of Aβ aggregation and neurotoxicity more effectively in in vitro and in vivo studies. In cellular and animal models of Alzheimer’s disease, we expected that the molecule trans-2-phenylvinyl-boronic-acid-MIDA-ester (TPVBA) and trans-beta-styryl-boronic-acid (TBSBA) minimized neuropathological abnormalities. They found that TBSBA reduced A42 agglomeration and enhanced cell viability more efficiently than TPVBA in a dot-blot experiment with cultivated N2a cells. The benefits of TBSBA were applied to C. elegans-expressing A42 and the 5xFAD mouse models of Alzheimer’s disease. Over two months, oral treatment of a 0.5 mg/kg dose of TBSA or an equal amount of methylcellulose vehicle to groups of six- and twelve-month-old 5xFAD or wild-type mice managed to prevent recognition and spatial-memory deficiencies within novel-object recognition and Morris water-maze storage tasks, respectively, and decreased the amount of nuclei and degenerated cells, A plaques, and GFAP and Iba-1 immunoreactivity in the cerebral cortex. 

Kucukdogru et al. [228] studied boron nitride NP-based systems in the Parkinson’s disease model to prevent 1-methyl-4-phenylpyridinium (MPP+)-induced apoptosis. In an experimental Parkinson’s disease model, ameliorative effects of BN NPs against toxicity of MPP+ were investigated. MPP+ was used to differentiate pluripotent human embryonal carcinoma cell (Ntera-2, NT-2) growth in a broad range of doses to generate an experimental Parkinson’s disease model (0.62 to 2 mM). Cell viability measurements such as MTT and LDH release were used to investigate the neuroprotective efficacy of BN NPs toward MPP+ toxicity. Antioxidant capacity (AC) and oxidant status (OS) analyses were used to study oxidative changes caused by the application of BN NPs in a Parkinson’s disease tissue culture model. Using the Hoechst 33258 fluorescent staining technique, the effects of BN NPs and MPP+ on nuclear stability were evaluated. A colorimetric assay was used to assess acetylcholinesterase enzyme activity in response to BN NPs treatment. Flow cytometry analysis was utilized to examine the processes of cell death caused by BN NPs and MPP+ exposure. The application of BN NPs enhanced cellular uptake in a Parkinson’s disease model as compared to the application of MPP+. Antioxidant capacity increased upon BN NPs applications, while oxidant levels decreased, according to AS and OS analyses. Moreover, flow cytometry analysis revealed that after treatment with BN NPs, MPP+-induced apoptosis was significantly reduced (*p* < 0.05). Zhu et al. [229] introduced functionalized boron-based nanoparticles as impending capable antimalarial agents. A cascade approach was used to synthesize boron nanoparticles (BNPs) functionalized using hydroxyl groups in situ, followed by bromination and hydroxylation processes. The synthesized BNPs retain low cytotoxicity and cell membrane penetrability. These nanoparticles were also investigated in vitro for antimalarial activity against Plasmodium falciparum (3D7 strain) and showed minimal side effects on Uppsala 87 malignant glioma (U87MG) cells, malignant melanoma A375 cells, KB human oral tumor cells, rat cortical neuron cells, and rat fibroblast-like synoviocyte cells with an IC50 value of 0.0021 µM. In comparison with current therapeutic antimalarial drugs such as pyrimethamine and chloroquine, BNPs exhibit higher antimalarial efficacy in in vitro. BNPs seem to have a strong promise for the synthesis of next-generation antimalarial drugs along with nanomaterial-based drugs for malaria.

Garcia et al. [230] reported light-responsive BODIPY-based quinolone (Figure 12) release for antimicrobial activity (**16**–**18**). To protect two separate quinolone-based molecules and inactive their antibacterial properties, we utilized BODIPY fluorophore with significant visual absorption. The BODIPY fluorophore has been connected to the quinolones at position 3, which is necessary for their antibacterial activity, and it can be resumed by exposing it to green or red light. These compounds were essentially soluble in biologically inert water and DMSO solutions, allowing us to investigate antibacterial characteristics before and after light irradiation. The photocage has a significant deactivation effect on the molecule’s activity, with better studies revealing a 32-fold increase in activity. A quinolone-susceptible *E*. *coli* strain was used to test the activity of these compounds. Andersen et al. [231] described a ROS-responsive boron-based prodrug for the delivery of methotrexate to inflammatory tissues (**19**, **20**). These drug candidates demonstrated good chemical and metabolic stability in blood, acidic or basic conditions, and liver microsomes under varied physiological environments (Figure 12). In the CIA DBA/1J mouse model, equimolar dosages of drug candidates **19** (9.7 mg/kg) and **20** (8.8 mg/kg) given once every day over 2 weeks decreased the incidence of arthritis, with a similar outcome to those provided with MTX (7.0 mg/kg). The system treated with **19** and **20** showed no noticeable side effects, whereas the MTX-treated system lost a significant amount of weight. These findings suggested that the ROS-responsive prodrug approach, as demonstrated by prodrugs **19** and **20**, can be used to convert anti-inflammatory substances into prodrugs with long-term efficacy in preventing the progression of arthritic disease and enhanced safety profiles. Kumar et al. [232] studied a cancer theranostic drug **21** that responds to ROS and targets mitochondria for efficient treatment of cancer and precise in vivo cancer diagnosis (Figure 12). Theranostic prodrug **21** has four structural functional groups: (i) biotin moiety that targets cancer and directs **21** to cancer tissues; (ii) fluorescent reporter that targets the mitochondria and in physiological aqueous conditions, ethidium exhibits a modest fluorescent signal, but when it interacts with double-stranded DNA or RNA, it becomes intensely fluorescent; (iii) inactive drug 5′-deoxy-5-fluorouridine (5′-DFU), which when hydrolyzed by abundantly expressed thymidine phosphorylase in tumor cells, becomes the active form, 5-fluorouracil; and (iv) self-immolating linker joined to a ROS-responsive phenyl boronate moiety. The parent anticancer drug 5′-DFU is released from the phenyl boronate component when it is exposed to increased ROS in cancer tissues. Next, the self-immolating linkage is severed, releasing the activated fluorescent reporter with a potent signal at 590 nm. In vitro, cell-based experiments demonstrated that **21**, while showing no noticeable cytotoxicity against normal cells, seemed to have greatly improved anticancer activity and sensitivity and selectivity targeting cancerous cells with overexpressed biotin transporters and higher ROS levels. Theranostic prodrug **21** was able to be distributed selectively in the tumor tissue, showing a fluorescence intensity that was roughly seven times higher than other healthy cells, including the respiratory system, liver, cardiovascular system, kidney, and central nervous system, in in vivo experiments using a xenograft mouse model inoculated with human A549 lung cancer cells. After the eight-week treatment period in vivo, **21** had a strong inhibitory effect on tumor growth.

## 9. Boron-Based Materials in Other Applications

Bioactive glasses have been extensively explored as biomaterials for precise uses in healthcare, dental care, pharmacy, dermatology, and biochemistry. Owing to their unique bone-forming biocompatibility, proteolysis capacity, and desirable physiological activities of their reactive species on osteoblastic activities, bioactive glasses (BGs) have been extensively used as platforms for tissue engineering applications as well as for in situ tissue regeneration [233]. In addition, altering the BGs’ chemical makeup by adding or removing various ions may control their bioactivity. Boron is a significant element that has been thought to be important for bone pathophysiology. It has been observed that a boron shortage has led to the change or loss of crucial physiological processes related to the metabolic activities of calcium and the development and rebuilding of bone tissue [234,235]. In addition, recent research has demonstrated that boron inclusion in BGs might promote osteogenesis in both in vitro and in vivo [236]. Angiogenesis may be essential for bone tissue regeneration in vivo, according to some findings [237]. This section focuses on bioactive glasses as potential biomaterials for regenerative medicine. 

Xia et al. [238] reported boron-based bioactive glass for bone regeneration using polycaprolactone. There has been a large amount of interest in polycaprolactone (PCL) for bone regeneration due to its cost-effectiveness, durability, and high degradability. Pure PCL’s weak bioactivity, hydrophobicity, and poor deformability has prevented it from being used more widely for bone regeneration. In order to assess the prospective uses of boron–BG/PCL materials for bone regeneration, the effects of boron–BG components on the physical characteristics and biological function of PCL polymer were evaluated at varied boron–BG quantities (0, 10, 20, 30, and 40 wt%). The findings show that as compared to standard PCL polymers, boron–BG/PCL composite materials have improved mechanical properties, a human optimum pH value, and quick breakdown. Furthermore, when compared to standard PCL polymers, the addition of boron–BG can promote the growth, differentiation into osteoblasts, and expression of angiogenic factors in rat stromal cells of bone marrow. Boron–BG was significant in that it enhanced angiogenic development in endothelial cells from human umbilical veins. The level of boron–BG had an impact on these improved effects, with a 30 weight percentage of boron–BG/PCL composite materials having the strongest enhancing effect. The 30 weight percentage of boron–BG/PCL composite materials could therefore be used in bone regeneration fields. Zheng et al. [239] described porous BG NPs containing boron to minimize inflammatory responses and to postpone osteogenic progression. Mesoporous BG NPs are versatile main components used in nanomedicine and tissue regeneration. Ions that are bioactive can give mesoporous BG NPs unique capabilities for enhanced therapeutic effects. By utilizing a sol–gel technique, boron is added to mesoporous BG NPs. By adjusting the quantity of the boron precursor, one may regulate the percentage of boron integrated. It is possible to create two different forms of boron-doped mesoporous BG NPs, known as 10B- and 15B- mesoporous BG NPs (5.8 and 6.5 mol% of B_2_O_3_, respectively). The inclusion of boron has no discernible impact on the particle shape. The size range of the produced particles is between 100 and 300 nm, and they all have a sphere-like form. Significant specific surface areas (346 and 320 m^2^ g^−1^, respectively) and pore volumes are displayed by 10B- and 15B mesoporous BG NPs. Both types of boron-doped mesoporous BG NPs have exceptional in vitro biocompatibility and are not cytotoxic. By regulating the levels of pro-inflammatory proteins, boron inclusion has been found to lower the inflammatory response associated with macrophages. The activity of pro-osteogenic genes is downregulated, which suggests that boron inclusion slows osteogenic development in osteoblasts. Utilizing boron-doped mesoporous BG NPs to modify the inflammatory response for bone tissue regeneration during inflammatory circumstances has been proven for the first time in the current study, and the study reveals that this approach has a bright future.

Cal et al. [240] studied silica-based boron containing cryogels with a possible bone development ability. The cost-effective synthesis of silicon- and boron-incorporated sol–gel-enhanced collagen/hair keratin cryogels reactions. Tetraethyl orthosilicate was used as a silica precursor to interact with collagen and hair keratin with a B-Si network. Collagen and keratin, two biopolymer chains, are covalently bound together in the design method to create an organic–inorganic structure. The resulting cryogels also were investigated systematically using a variety of techniques. Comparing the generated cryogel to collagen/keratin bioscaffolds made using a conventional method, the latter showed enhanced mechanical properties and great thermophysical stability. The organic/inorganic cryogen also helped hAMSCs differentiate into osteoblasts. The inorganic component of the cryogen, which offers an extracellular matrix resembling bone tissue and eventually encourages cell initiation and tissue repair, may be responsible for this capacity.

## 10. Conclusions

In this review, we comprehensively discussed different types of boron-based materials such as boron nitride, boronic acid, and BODIPY in drug delivery for different diseases, which is either an exogenous or endogenous stimuli-responsive release. We have particularly focused on nanoscale-based drug delivery systems, such as nanoparticles, nanosheets/nanospheres, nanotubes, nanocarriers, microneedles, nanocapsules, hydrogel, nanoassembly, etc. This review also delivers the concept of designing a strategy and ability to attain controlled drug release over the response to particular stimuli such as pH, light, GSH, glucose, or temperature. Regarding boron-based drug delivery systems used as a particular drug carrier or drug moiety, as well as other essential drug delivery system design requirements, we have attempted to give readers a detailed understanding of these systems. The difficulties with the current boron-based materials for drug delivery systems are projected to be addressed in this review, and new multi-stimuli responsive boron-based materials for drug delivery are expected to be developed.

## Data Availability

Not applicable.

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
