# Peer review of "Stimuli-Responsive Boron-Based Materials in Drug Delivery"

_ijms, 2023, doi:10.3390/ijms24032757_

Round 1

Reviewer 1 Report

In this manuscript, Das and co-workers did a nice job reviewing recent developments on boron-containing materials employed in drug delivery applications. They mainly focused on three types of boron materials, including boron nitride (BN), boronic acid, and BODIPY. The background is sufficiently introduced, and the representative examples are well-reviewed. Drug delivery systems (DDS) have emerged as highly effective strategies and have numerous advantages over free drugs. This review could be a nice addition to the field; thus, the reviewer recommends the publication of the manuscript after some minor edits. 

1. The authors did a short review of different drug delivery methods in section 2, which is nice. However, the reviewer recommends the authors add refs to this section, which could be beneficial for the readers if they want to look deeper into this. 

2. For boronic acid materials, other than polymeric materials, boronic acid-containing liposomes have also received a lot of attention over the past several years (i.e., Am. Chem. Soc., 1998, 120, 7141–7142Chem. Commun., 2000, 149–150Chem. Commun., 2018, 54, 6169-6172, ChemBioChem 2022, 23, e202200402, etc.). The authors could think about expanding the section a bit more by adding these examples. 

Author Response

Response to Reviewer 1:

In this manuscript, Das and co-workers did a nice job reviewing recent developments on boron-containing materials employed in drug delivery applications. They mainly focused on three types of boron materials, including boron nitride (BN), boronic acid, and BODIPY. The background is sufficiently introduced, and the representative examples are well-reviewed. Drug delivery systems (DDS) have emerged as highly effective strategies and have numerous advantages over free drugs. This review could be a nice addition to the field; thus, the reviewer recommends the publication of the manuscript after some minor edits. 

  1. 1. The authors did a short review of different drug delivery methods in section 2, which is nice. However, the reviewer recommends the authors add refs to this section, which could be beneficial for the readers if they want to look deeper into this. 

Response: Thank you for your suggestions, we have included the necessary references in the revised manuscript (Page No: 5,6 & Line 158-213) and the new reference number is 114-124.

  1. 2. For boronic acid materials, other than polymeric materials, boronic acid-containing liposomes have also received a lot of attention over the past several years (i.e.,Am. Chem. Soc., 1998, 120, 7141–7142, Chem. Commun., 2000, 149–150, Chem. Commun., 2018, 54, 6169-6172, ChemBioChem 2022, 23, e202200402, etc.). The authors could think about expanding the section a bit more by adding these examples. 

Response: We have expanded the section according to the reviewer’s comments (Page No: 21,22 & Line 909-954) and added references and the new reference number is 211-214.

Reviewer 2 Report

Bhaskar C. Das and his team reviewed a topic entitled “Stimuli-Responsive Boron-Based Materials in Drug Delivery.” The authors focused on the recent progress of bioenvironmental-responsive Boron-containing compounds and polymers for drug delivery applications.                                                                                                          

Overall, the article is well-framed and worth publishing in the MDPI International Journal of Molecular Sciences. However, I recommend a detailed revision addressing the following major and minor issues carefully to reach out broader audience of bioengineering, biology, and materials science before considering a possible publication.

Major issues:

1.      One of the weaknesses of this review is unnecessary information and discussion, which is not directly related to the main topic of boron-based materials.                  For example,                                                                                                                               The authors introduced structures of anticancer drugs in Figure 3. None of them contain any boron atom. Note that this review is not about anticancer drugs. Similarly, no information or discussion about boron-related compounds and their administration was provided in the section on “Types of Drug Delivery Methods” and Figure 2.  On the same token, how “Figure 4Cell cycle showing the different phases of cell division” is connected with Boron-compounds, out of the mainstream topic, i.e., stimuli-responsive boron materials in drug delivery.

2.      Figure 1: Show the “N” atom of aza-BODIPY in a different color, which will help the readers quickly identify the difference between BODIPY and aza-BODIPY. 

3.      The authors failed to show metal in Figure 8. Structure of MOF-based BODIPY. The authors provided only BODIPY-derivate bipyridine (BBP). 

4.      Figures 8, 9, 10, 11, 12, 13, 14, 15, 16, and 17 can be merged into either one or two.

5.      Specify the advantages of boronic acids as stimuli-responsive functional groups compared to other bio-responsive groups such as thiols (-SH). 

For example, boronic acid groups respond to various chemicals/metabolites of the disease site, such as reactive oxygen species, ATP, glucose, and acidic pH. In addition, boronic acid groups also act as ligands for targeting diseased cells. For instance, abundant expression of sialic acid on cell membranes of cancer cells enables boronic acid groups to target the cancer cells (Acc Chem Res 2019;52(11):3108-3119 and Angew Chem Int Ed Engl 2010;49(32):5494-7).

6.      I strongly recommend that the authors revise section “6. Boronic acid-based Drug Delivery”, which is the heart of the review. The following articles will help the authors to emphasize the role of stimuli-sensitive boron-based materials in modulating the intracellular release kinetics of different drugs (mRNA, pDNA, siRNA, and drugs) from the nanocarriers. Messenger RNA delivery (J Control Release 2021;330:317-328 and Adv Healthc Mater 2022;11(9):e2102016). pDNA delivery (J Am Chem Soc 2017;139(51):18567-18575). siRNA delivery (Angew Chem Int Ed Engl 2012;51(43):10751-5).

7.      Phenylboronic esters were employed to construct the stimuli-sensitive prodrugs. I recommend that the authors introduce this scientific information in the manuscript. Self-destruction of phenylboronic ester groups in nanoreactors releases quinone methide (QM) as a by-product in response to high levels of hydrogen peroxide produced in the tumor. The released QM depletes glutathione and suppresses the antioxidant ability of tumor cells, thereby causing efficient cancer-killing (Angew Chem Int Ed Engl 2017;56(45):14025-14030). 

Minor issues

1.      Redundant acronyms were used. 

For example: 

Line 89, Line 704: phenylboronic acid (PBA)) 

Line 20, Line 63, Line 440, and Line 1209: boron nitride (BN) 

Line 489 and Line 1129: 5-fluorouracil (5-FU) 

Line 835 and Table 1 footnotes, Line Nanoparticles (NPs) 

Line 810, Boron-dipyrromethene (BODIPY) 

Line 579, Line 869: doxorubicin (DOX) 

Line 41, Line 587: drug delivery system (DDS)

Line 849, Line 1108: methotrexate (MTX) 

2.      Acronyms must be defined after their first appearance in the manuscript. 

Line 102 and Line 592: Give the acronym ROS in Line 102 after reactive oxygen species (ROS)Remove reactive oxygen species in Line 592 and use ROS.

3.      Take care of the appropriate use of Periods. 

For example.

3.1 . Polymers ïƒ   3.1. Polymers

3.2 . Nanoparticles ïƒ  3.2. Nanoparticles 

3.3 . Nanocapsules ïƒ  3.3. Nanocapsules 

3.4 . Nanotubes ïƒ  3.4. Nanotubes 

3.5 . Nanogels ïƒ  3.5. Nanogels 

3.6 . Dendrimers ïƒ  3.6. Dendrimers 

10. . Conclusions ïƒ  10. Conclusions

4. Stereochemical information should be italicized.

For example:

Line 113: 4,4-Difluoro-4-bora-3a,4a-diaza-s-indacene ïƒ  4,4-Difluoro-4-bora-3a,4a-diaza-s-indacene

Subscripts and superscripts should be properly used. 

Line 123: CH2 group ïƒ  CH2 group

Line 615: >2 g L-1 ïƒ  >2 g L-1

Line 674: 0.48 m-1 ïƒ  0.48 m-1

Line 824: H2S ïƒ  H2S

5.  Acronyms or abbreviations, or chemical formulas are not required if the information is given only once. These will distract the readers. 

For example:

Line 446: molybdenum disulfide (MoS2) ïƒ  molybdenum disulfide (MoS2), 

Line 446: tungsten disulfide (WS2) ïƒ  tungsten disulfide (WS2

Line 447: Bismuth selenide (Bi2Se3) ïƒ  Bismuth selenide (Bi2Se3

Line 453: epoxy (-O) 

Line 489:  6-mercaptopurine (6-MP)

Line 512 and 531: folic acid (FA)

Line 579: CM

Line 580: (BNNTs-CM-DOX) 

6.  Define the following words:

Line 487 to Line 499: post mitotic gap phase (G1 phase), synthesis phase (S phase), Gap 2 (G2 phase), and mitotic phase (M phase)

7.          It is better to remove acronyms or abbreviations in the abstract. Expand the full form of acronyms or abbreviations in the abstract. 

For example, boron nitride (BN)

GSH ïƒ  glutathione

8.          Expand the following.

GSH: glutathione

BODIPY: Boron dipyrromethene

9.  Unnecessary capital letters were observed in the middle of the sentences. 

For example: 

Line 97: the Nucleophilic combination ïƒ  the nucleophilic combination

Line 407: For example, Dehydration condensation ïƒ  For example, dehydration condensation 

Author Response

Response to Reviewer 2:

Overall Comment: Bhaskar C. Das and his team reviewed a topic entitled “Stimuli-Responsive Boron-Based Materials in Drug Delivery.” The authors focused on the recent progress of bioenvironmental-responsive Boron-containing compounds and polymers for drug delivery applications. Overall, the article is well-framed and worth publishing in the MDPI International Journal of Molecular Sciences. However, I recommend a detailed revision addressing the following major and minor issues carefully to reach out broader audience of bioengineering, biology, and materials science before considering a possible publication.

Response: We are heartily thankful to reviewer 2 for valuable suggestions and accepting our article. We addressed all major and minor issues raised by reviewer in major manuscript and detailed are reported here.

Major issues:

  1. 1.One of the weaknesses of this review is unnecessary information and discussion, which is not directly related to the main topic of boron-based materials. For example, the authors introduced structures of anticancer drugs in Figure 3. None of them contain any boron atom. Note that this review is not about anticancer drugs. Similarly, no information or discussion about boron-related compounds and their administration was provided in the section on “Types of Drug Delivery Methods” and Figure 2.  On the same token, how “Figure 4. Cell cycle showing the different phases of cell division” is connected with Boron-compounds, out of the mainstream topic, i.e., stimuli-responsive boron materials in drug delivery.

Response 1:

  • Thank you for your concern, we appreciate your thoughts. As our review article focus on boron-based material for drug delivery mainly (anticancer drug), So we added types of available anticancer drugs and how materials help in delivery to release these drugs in stimuli environment was shown in 3.  This will help readers to  have some ideas about the anticancer drugs.
  • Similarly, we have discussed only the generally available drug delivery methods in section 2 and it is shown in 2. Most of boron materials-based drug delivery systems are oral-based delivery. And compared to oral, other types of drug delivery methods have some advantages, but hardly used boron materials. Boron based materials to be used in different types of drug delivery system is very frontier areas of research. May be in the near future other researchers will focus in this area. The advantages of each method are explained in this section. But we focused on only oral drug  delivery. Our review article’s aim is to focus on boron based materials in oral drug delivery system. We have explained detailed in Page No. 7 & Line 214-218.
  • In 4, we have discussed the importance of phase-selective drugs (anticancer drugs). It is very important to select drugs according to their phase-specific characteristics to urge more G0 phase cells to enter the proliferating cycle to increase the number of tumor cells killed by drugs. Such phase-specific anticancer treatment has been explained schematically in Fig. 4.
  1. 2.Figure 1: Show the “N” atom of aza-BODIPY in a different color, which will help the readers quickly identify the difference between BODIPY and aza-BODIPY. 

Response 2: We have modified Fig.1 as per your suggestion and it is included in the revised manuscript (Page No: 3)

  1. 3.The authors failed to show metal in Figure 8. Structure of MOF-based BODIPY. The authors provided only BODIPY-derivate bipyridine (BBP). 

Response 3: Thank you for your suggestions, we have added the metal Fig. 8 (Page No: 24) and it is included in the revised manuscript.

  1. 4.Figures 8, 9, 10, 11, 12, 13, 14, 15, 16, and 17 can be merged into either one or two.

Response 4: Thank you for your suggestions, we have merged them into two figures and it is included in the revised manuscript (Page No: 27 & 32).  New Figure 10 and 11.

  1. 5.Specify the advantages of boronic acids as stimuli-responsive functional groups compared to other bio-responsive groups such as thiols (-SH). For example, boronic acid groups respond to various chemicals/metabolites of the disease site, such as reactive oxygen species, ATP, glucose, and acidic pH. In addition, boronic acid groups also act as ligands for targeting diseased cells. For instance, abundant expression of sialic acid on cell membranes of cancer cells enables boronic acid groups to target the cancer cells (Acc Chem Res 2019;52(11):3108-3119 and Angew Chem Int Ed Engl 2010;49(32):5494-7).

Response 5: We appreciate your suggestions; we have included the advantages of boronic acids as stimuli-responsive groups in section 6 (Page No: 15,16 & Line 615-636) and added references and the new reference number is 198-199. Using BA as a drug-conjugated linker offers a useful method for managing the release of drugs in response to certain ROS concentrations associated with disease activity.

  1. 6.I strongly recommend that the authors revise section “6. Boronic acid-based Drug Delivery”, which is the heart of the review. The following articles will help the authors to emphasize the role of stimuli-sensitive boron-based materials in modulating the intracellular release kinetics of different drugs (mRNA, pDNA, siRNA, and drugs) from the nanocarriers. Messenger RNA delivery (J Control Release 2021;330:317-328 and Adv Healthc Mater 2022;11(9):e2102016). pDNA delivery (J Am Chem Soc 2017;139(51):18567-18575). siRNA delivery (Angew Chem Int Ed Engl 2012;51(43):10751-5).

Response 6: We are thankful to you for your  suggestions, we have revised section 6 (Page No: 20, 21 & Line 836-908) and added references and the new reference number is 207-210. We have elaborately discussed mRNA, pDNA, and siRNA releases in this section.

  1. 7.Phenylboronic esters were employed to construct the stimuli-sensitive prodrugs. I recommend that the authors introduce this scientific information in the manuscript. Self-destruction of phenylboronic ester groups in nanoreactors releases quinone methide (QM) as a by-product in response to high levels of hydrogen peroxide produced in the tumor. The released QM depletes glutathione and suppresses the antioxidant ability of tumor cells, thereby causing efficient cancer-killing (Angew Chem Int Ed Engl 2017;56(45):14025-14030). 

Response 7: Thanks a lot for your suggestion, we have included QM importance in section 6 (Page No: 22 & Line 955-971), the oxidation reaction is catalyzed to produce enormous amounts of H2O2 by the presence of glucose oxidase. The chronic stress in the tumor site rises in tandem with the decrease in nutritional material content. In contrast, the high H2O2 level damages PPBEM segments and causes the vesicles to self-destruct, releasing quinone methide (QM) as a residue. The capacity of QM to reduce intracellular GSH affects the capacity of cancer cells to resist oxidative stress. Raising oxidative stress and decreasing GSH work together to effectively destroy cancer cells and stop the expansion of tumors. The new reference number is 215.

Minor issues:

  1. 1.Redundant acronyms were used. 

For example: 

Line 89, Line 704: phenylboronic acid (PBA)) 

Line 20, Line 63, Line 440, and Line 1209: boron nitride (BN) 

Line 489 and Line 1129: 5-fluorouracil (5-FU) 

Line 835 and Table 1 footnotes, Line Nanoparticles (NPs) 

Line 810, Boron-dipyrromethene (BODIPY) 

Line 579, Line 869: doxorubicin (DOX) 

Line 41, Line 587: drug delivery system (DDS)

Line 849, Line 1108: methotrexate (MTX) 

Response 1: We have corrected all the redundant acronyms and all abbreviations included at  the top of the references section and it is included in the revised manuscript (Page No. 35 & Line 1386-1396)

Abbreviations

            DDS: Drug delivery system; BN: Boron nitride; BODIPY: Boron dipyrromethene; PBA: Phenyl boronic acid; ATP: Adenosine triphosphate; ROS: Reactive oxygen species; NSs: Nanosheets/ Nanospheres; NTs: Nanotubes; GSH: Glutathione; MS: Mesoporous silica; NCs: Nanocarriers;, NPs: Nanoparticles; MNs: Microneedle, NCPs: Nanocapsules; Hy: Hydrogel; MOF: Met-al-organic frameworks; DLNP: Dendrimer-based lipid nanoparticle; NA: Nanoassembly; GI: Gastro; BNNSs: Boron nitride nanosheets; GF: Growth fraction; DOX: Doxorubicin; CPT: Camptothecin; BNNTs: Boron nitride nanotubes; H2O2: Hydrogen peroxide; BA: Boronic acid; PEG: poly ethylene glycol; RAFT: Reversible addition fragmentation chain transfer; SF: Silk fibroin; PM: Polyplex micelles; FPBA: 4-carboxy-3-fluorophenylboronic acid; PIC: Polyion complex; NRs: Nanoreactors; GOD: glucose oxidase; QM: Quinone methide; PDT: Photodynamic therapy; FRET: Fluorescence resonance energy transfer; DTX: Docetaxel; AD: Alzheimer's disease; BBB: blood-brain barrier; TBSBA: Trans-beta-styryl-boronic-acid; MPP+: 1-methyl-4-phenylpyridinium; AC: Antioxidant capacity; OS: Oxidant status; 5′-DFU: 5′-deoxy-5- fluorouridine; BGs: Bioactive glasses; PCL: Polycaprolactone.

  1. 2.Acronyms must be defined after their first appearance in the manuscript. 

Line 102 and Line 592: Give the acronym ROS in Line 102 after reactive oxygen species (ROS). Remove reactive oxygen species in Line 592 and use ROS.

Response 2: We have corrected the ROS mistakes in the revised manuscript.

  1. 3.Take care of the appropriate use of Periods. 

For example.

3.1 . Polymers à  3.1. Polymers

3.2 . Nanoparticles à 3.2. Nanoparticles 

3.3 . Nanocapsules à 3.3. Nanocapsules 

3.4 . Nanotubes à 3.4. Nanotubes 

3.5 . Nanogels à 3.5. Nanogels 

3.6 . Dendrimers à 3.6. Dendrimers 

  1. . Conclusions à10. Conclusions

Response 3: We have corrected the Periods errors in the revised manuscript.

  1. 4.Stereochemical information should be italicized.

For example:

Line 113: 4,4-Difluoro-4-bora-3a,4a-diaza-s-indacene à 4,4-Difluoro-4-bora-3a,4a-diaza-s-indacene

Subscripts and superscripts should be properly used. 

Line 123: CH2 group à CH2 group

Line 615: >2 g L-1 à >2 g L-1

Line 674: 0.48 m-1 à 0.48 m-1

Line 824: H2S à H2S

 Response 4: We have corrected the Stereochemical, Subscripts, and Superscripts errors in the revised manuscript.

  1. 5.Acronyms or abbreviations, or chemical formulasare not required if the information is given only once. These will distract the readers. 

For example:

Line 446: molybdenum disulfide (MoS2) à molybdenum disulfide (MoS2), 

Line 446: tungsten disulfide (WS2) à tungsten disulfide (WS2

Line 447: Bismuth selenide (Bi2Se3) à Bismuth selenide (Bi2Se3

Line 453: epoxy (-O) 

Line 489:  6-mercaptopurine (6-MP)

Line 512 and 531: folic acid (FA)

Line 579: CM

Line 580: (BNNTs-CM-DOX) 

Response 5: We have corrected the Acronyms, abbreviations, and chemical formula errors in the revised manuscript.

  1. 6.Define the following words:

Line 487 to Line 499: post mitotic gap phase (G1 phase), synthesis phase (S phase), Gap 2 (G2 phase), and mitotic phase (M phase)

Response 6 We have included the definition in the revised manuscript (Page No.13 & Line 497-501).

  1. 7.It is better to remove acronyms or abbreviations in the abstract. Expand the full form of acronyms or abbreviations in the abstract. 

For example, boron nitride (BN)

GSH à glutathione

Response 7: As per your suggestions, we have corrected the abstract section and included it in the revised manuscript.

  1. 8.Expand the following.

GSH: glutathione

BODIPY: Boron dipyrromethene

Response 8: We have corrected the mistakes in the revised manuscript.

  1. 9.Unnecessary capital letters were observed in the middle of the sentences. 

For example: 

Line 97: the Nucleophilic combination à the nucleophilic combination

Line 407: For example, Dehydration condensation à For example, dehydration condensation 

Response 9: We have corrected the capital letter mistakes

Round 2

Reviewer 2 Report

Accepted in present form